# Structural basis for the absence of low-energy chlorophylls in a photosystem I trimer from *Gloeobacter violaceus*

**Koji Kato**[1†], **Tasuku Hamaguchi**[2†], **Ryo Nagao**[1*†], **Keisuke Kawakami**[2], **Yoshifumi Ueno**[3], **Takehiro Suzuki**[4], **Hiroko Uchida**[5], **Akio Murakami**[3,5], **Yoshiki Nakajima**[1], **Makio Yokono**[6], **Seiji Akimoto**[3], **Naoshi Dohmae**[4], **Koji Yonekura**[2,7,8*], **Jian-Ren Shen**[1*]

[1]Research Institute for Interdisciplinary Science and Graduate School of Natural Science and Technology, Okayama University, Okayama, Japan; [2]Biostructural Mechanism Laboratory, RIKEN SPring-8 Center, Hyogo, Japan; [3]Graduate School of Science, Kobe University, Hyogo, Japan; [4]Biomolecular Characterization Unit, RIKEN Center for Sustainable Resource Science, Saitama, Japan; [5]Research Center for Inland Seas, Kobe University, Hyogo, Japan; [6]Institute of Low Temperature Science, Hokkaido University, Hokkaido, Japan; [7]Institute of Multidisciplinary Research for Advanced Materials, Tohoku University, Miyagi, Japan; [8]Advanced Electron Microscope Development Unit, RIKEN-JEOL Collaboration Center, RIKEN Baton Zone Program, Hyogo, Japan

**\*For correspondence:**
nagaoryo@okayama-u.ac.jp (RN);
yone@spring8.or.jp (KY);
shen@cc.okayama-u.ac.jp (J-RenS)

[†]These authors contributed equally to this work

**Competing interest:** The authors declare that no competing interests exist.

**Abstract** Photosystem I (PSI) is a multi-subunit pigment-protein complex that functions in light-harvesting and photochemical charge-separation reactions, followed by reduction of NADP to NADPH required for $CO_2$ fixation in photosynthetic organisms. PSI from different photosynthetic organisms has a variety of chlorophylls (Chls), some of which are at lower-energy levels than its reaction center P700, a special pair of Chls, and are called low-energy Chls. However, the sites of low-energy Chls are still under debate. Here, we solved a 2.04-Å resolution structure of a PSI trimer by cryo-electron microscopy from a primordial cyanobacterium *Gloeobacter violaceus* PCC 7421, which has no low-energy Chls. The structure shows the absence of some subunits commonly found in other cyanobacteria, confirming the primordial nature of this cyanobacterium. Comparison with the known structures of PSI from other cyanobacteria and eukaryotic organisms reveals that one dimeric and one trimeric Chls are lacking in the *Gloeobacter* PSI. The dimeric and trimeric Chls are named Low1 and Low2, respectively. Low2 is missing in some cyanobacterial and eukaryotic PSIs, whereas Low1 is absent only in *Gloeobacter*. These findings provide insights into not only the identity of low-energy Chls in PSI, but also the evolutionary changes of low-energy Chls in oxyphototrophs.

## Editor's evaluation

This work presents a high-resolution structure of a photosystem I complex of the primordial cyanobacterium *Gloeobacter violaceus*. It relates the structural differences with oher cyanobacteria to differences in the optical properties.

## Introduction

Oxygenic photosynthetic reactions convert light energy into chemical energy and produce molecular oxygen, thereby maintaining the aerobic life on the earth (*Blankenship, 2021*). The light-induced photosynthetic reactions occur in two multisubunit pigment–protein complexes, photosystem I and photosystem II (PSI and PSII, respectively). PSI functions in light-harvesting, charge-separation, and electron-transfer reactions, leading to the reduction of NADP to NADPH required for $CO_2$ fixation (*Blankenship, 2021*; *Brettel and Leibl, 2001*; *Nelson and Yocum, 2006*). PSI contains many cofactors, including chlorophylls (Chls), carotenoids (Cars), quinones, and iron–sulfur clusters, which serve in the photochemical reactions (*Nelson and Yocum, 2006*). These cofactors are well conserved in PSI cores among oxyphototrophs, although the oligomerization states and subunit compositions of PSI differ significantly among different species of organisms (*Fromme et al., 2001*; *Suga and Shen, 2020*; *Hippler and Nelson, 2021*).

Among photosynthetic organisms, cyanobacteria are a large group of prokaryotes, and their PSI cores are present in either trimeric (*Jordan et al., 2001*; *Malavath et al., 2018*; *Kato et al., 2020*; *Gisriel et al., 2020*; *Hamaguchi et al., 2021*; *Xu et al., 2021*) or tetrameric (*Kato et al., 2019*; *Zheng et al., 2019*; *Chen et al., 2020*; *Semchonok et al., 2022*) forms. Cyanobacterial PSI complexes also possess unique Chl molecules at energy levels lower than the level of P700, a special pair Chl that performs charge separation in each PSI-monomer unit (*Brettel and Leibl, 2001*). The low-energy Chls are historically called red Chls, which differ in their energy levels among different species of cyanobacteria (*Gobets and van Grondelle, 2001*; *Schlodder et al., 2005*; *Karapetyan et al., 2006*; *Schlodder et al., 2007*). The main functions of low-energy Chls in PSI are either uphill energy transfer from low-energy Chls to other Chls (*Gobets and van Grondelle, 2001*) or excitation-energy quenching upon P700 oxidation (*Shubin et al., 1995*; *Shibata et al., 2010*; *Schlodder et al., 2011*). From these observations, the low-energy Chls are thought to regulate energy balance for energy transfer and energy quenching (*Gobets and van Grondelle, 2001*; *Schlodder et al., 2005*; *Karapetyan et al., 2006*; *Schlodder et al., 2007*). Despite numerous structural and functional analyses of the cyanobacterial PSI, the locations of low-energy Chls are still under debate both experimentally and theoretically (*Gobets and van Grondelle, 2001*; *Schlodder et al., 2005*; *Karapetyan et al., 2006*; *Schlodder et al., 2007*; *Byrdin et al., 2002*; *Adolphs et al., 2010*). This is because it is very difficult to identify specific low-energy Chls from ~95 Chl molecules at different energy levels in a PSI monomer.

The sites of low-energy Chls in PSI have been suggested by spectroscopic techniques (*Gobets and van Grondelle, 2001*; *Schlodder et al., 2005*; *Karapetyan et al., 2006*; *Schlodder et al., 2007*). Absorption spectroscopy showed various peaks and shoulders around/over 700 nm, reflecting a complexity and difficulty of the identification of low-energy Chls in PSI. In contrast, fluorescence spectroscopy is a convenient method to observe low-energy Chls in PSI. It is known that under the liquid-nitrogen condition, two types of prominent fluorescence peaks from low-energy Chls were mainly found. These two fluorescence peaks appear at around 723 and 730 nm in PSI trimers isolated from the representative cyanobacteria, *Synechocystis* sp. PCC 6803 (hereafter referred to as *Synechocystis*) (*van der Lee et al., 1993*; *Mimuro et al., 2010*; *Turconi et al., 1996*) and *Thermosynechococcus elongatus* (*Pålsson et al., 1996*), respectively. The 723 and/or 730 nm fluorescence peaks are conserved in most cyanobacteria, although their band widths and peak positions differ slightly depending on the species of cyanobacteria as well as the experimental conditions employed to measure. The fluorescence feature of low-energy Chls is also conserved in the tetrameric PSI cores of cyanobacteria (*Kato et al., 2019*; *Nagao et al., 2020b*). However, it is unclear regarding the locations of these low-energy forms, because the PSI core contains about 95 Chls, and it is difficult to identify specific Chls from the bulk of the PSI Chls.

*Gloeobacter violaceus* PCC 7421 (hereafter referred to as *Gloeobacter*) is a unique cyanobacterium that has no thylakoid membranes, and its photosynthesis takes place in the cytoplasmic membranes similarly as that seen in anoxygenic photosynthetic bacteria (*Guglielmi et al., 1981*). Molecular phylogenetic analyses showed that *Gloeobacter* branched off from the main cyanobacterial tree at an early stage of evolution (*Nelissen et al., 1995*). Therefore, *Gloeobacter* is considered as an evolutionary primordial cyanobacterium. Unlike other cyanobacteria, the *Gloeobacter* PSI did not exhibit characteristic fluorescence peaks at around 723 or 730 nm in fluorescence-emission spectra both in vivo (*Koenig and Schmidt, 1995*) and in vitro (*Mimuro et al., 2010*). These observations lead to an

attractive notion that Chls absent in the structure of the *Gloeobacter* PSI are plausible candidates for Chls fluorescing at around 723 and 730 nm observed in other cyanobacteria.

To reveal the locations of characteristic low-energy Chls in PSI from *Synechocystis* and thermophilic cyanobacteria as well as the possible structural differences among PSI of *Gloeobacter* and other cyanobacteria, we solved the structure of the PSI trimer isolated from *Gloeobacter* using cryo-electron microscopy (cryo-EM) at a resolution of 2.04 Å. The structure obtained showed the absence of some subunits commonly found in other cyanobacteria, confirming the primordial nature of this cyanobacterium. By comparing the *Gloeobacter* PSI structure with the structures of *Synechocystis* and thermophilic cyanobacteria, we propose that one Chl dimer and one trimer, which are absent in the *Gloeobacter* PSI, are responsible for the characteristic fluorescence peaks at around 723 and 730 nm observed in other cyanobacteria.

## Results and discussion
### Overall structure of the PSI trimer

The PSI trimers were purified from *Gloeobacter*, and its biochemical characterization was summarized in *Appendix 1—figure 1*, which showed that the sample is very pure and suitable for structural analysis. Cryo-EM images of the PSI trimer were obtained by a JEOL CRYO ARM 300 electron microscope operated at 300 kV. After processing of the images with RELION (*Appendix 1—figure 2* and *Appendix 1—table 1*), the final cryo-EM map with a C3 symmetry enforced was determined at a resolution of 2.04 Å, based on the 'gold standard' Fourier shell correlation (FSC) = 0.143 criterion (*Figure 1A* and *Appendix 1—figure 2*). This resolution is the highest for the structure of PSI cores ever determined by X-ray crystallography and single-particle cryo-EM so far (*Jordan et al., 2001*; *Malavath et al., 2018*; *Kato et al., 2020*; *Gisriel et al., 2020*; *Hamaguchi et al., 2021*; *Xu et al., 2021*). This was realized by imaging with a cold-field emission electron beam that produces superior high-resolution signals (*Hamaguchi et al., 2019*). The atomic model of PSI was built based on the 2.04 Å map (*Figure 1B* and *Appendix 1—table 2*), and most of the cofactors and amino acid residues were precisely assigned to this high-resolution map.

The components of the electron-transfer chain are assigned unambiguously, which consists of a special pair Chls P700, accessory Chl Acc, primary electron acceptor $A_0$, secondary electron acceptor $A_1$ (menaquinone-4), and three iron–sulfur clusters $F_X/F_A/F_B$ (*Figure 2A*). The densities for the Mg atoms in the Chl molecules are clearly visualized at their centers (*Figure 2B–E*). In addition, the characteristic structures of Chl *a′* and Chl *a* that constitute the pair of P700 are distinguished clearly in the high-resolution map (red arrows in *Figure 2B, C*, respectively). The densities of individual heavy atoms in the iron–sulfur clusters are also clearly separated, allowing their precise assignment possible (*Figure 2G–I*). On the other hand, menaquinone-4 is assigned here (*Figure 2F*) because it has been detected in the *Gloeobacter* PSI core by HPLC (*Mimuro et al., 2005*), although it is difficult to distinguish between menaquinone-4 and phylloquinone at the present resolution of the cryo-EM map due to the same length of their prenyl chain. The electron-transfer chain is arranged in two branches approximately symmetrically, and no significant differences between the branches are found. Thus, it cannot be determined if the electrons are transferred through both branches or proceed preferentially through one branch based on the structure obtained.

### Subunit structures of the PSI monomer

PSI of *Gloeobacter* is a homo-trimeric complex, and its overall architecture is similar to those of the PSI trimer isolated from other cyanobacteria (*Jordan et al., 2001*; *Malavath et al., 2018*; *Kato et al., 2020*; *Gisriel et al., 2020*; *Hamaguchi et al., 2021*; *Xu et al., 2021*). Each monomer of the *Gloeobacter* PSI contains 10 subunits (*Figure 1B*), 9 of which are found in the genome of *Gloeobacter* (*psaA*, *psaB*, *psaC*, *psaD*, *psaE*, *psaF*, *psaL*, *psaM*, and *psaZ*) (*Nakamura et al., 2003*). PsaZ is positioned at the same location of PsaI in the PSI structure of *Synechocystis* (*Malavath et al., 2018*), although the *Gloeobacter* PsaZ has low sequence identity (20.0%) with the *Synechocystis* PsaI (*Appendix 1—figure 3A, B*; *Inoue et al., 2004*). The other eight subunits are located at similar positions of PSI from other cyanobacteria (*Jordan et al., 2001*; *Malavath et al., 2018*; *Kato et al., 2020*; *Gisriel et al., 2020*; *Hamaguchi et al., 2021*; *Xu et al., 2021*). An additional subunit is found at the position of PsaJ, which was modeled as polyalanines (*Appendix 1—figure 3C*), because the *psaJ* gene is not found

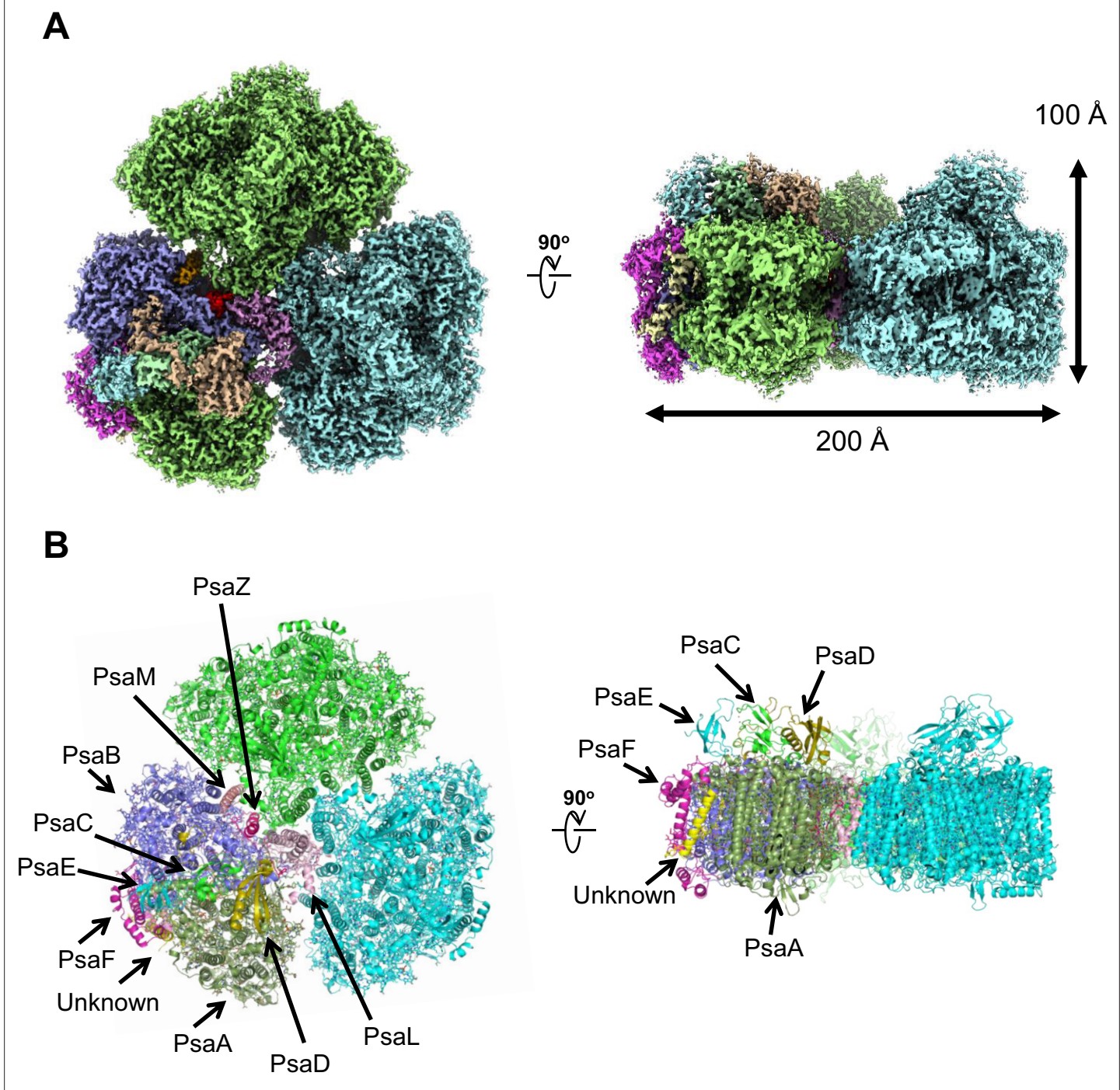

**Figure 1.** Overall structure of the *Gloeobacter* photosystem I (PSI) trimer. (**A**) Cryo-electron microscopy (cryo-EM) density of the PSI trimer at 2.04 Å resolution. (**B**) Structural model of the PSI trimer. Views are from the top of the cytosolic side (left) and side of the membrane (right) for both panels (**A** and **B**).

in the *Gloeobacter* genome (*Nakamura et al., 2003*) but a subunit is clearly visible in the map at the same position as PsaJ (*Jordan et al., 2001*; *Malavath et al., 2018*; *Kato et al., 2020*; *Gisriel et al., 2020*; *Hamaguchi et al., 2021*; *Xu et al., 2021*). This subunit is named Unknown in the *Gloeobacter* structure. The cryo-EM map of *Gloeobacter* PSI indicates the absence of PsaG, PsaH, PsaK, and PsaX found in other PSIs, consistent with the absence of their genes in the *Gloeobacter* genome (*Nakamura et al., 2003*). While PsaG and PsaH are also absent in other cyanobacteria and appear

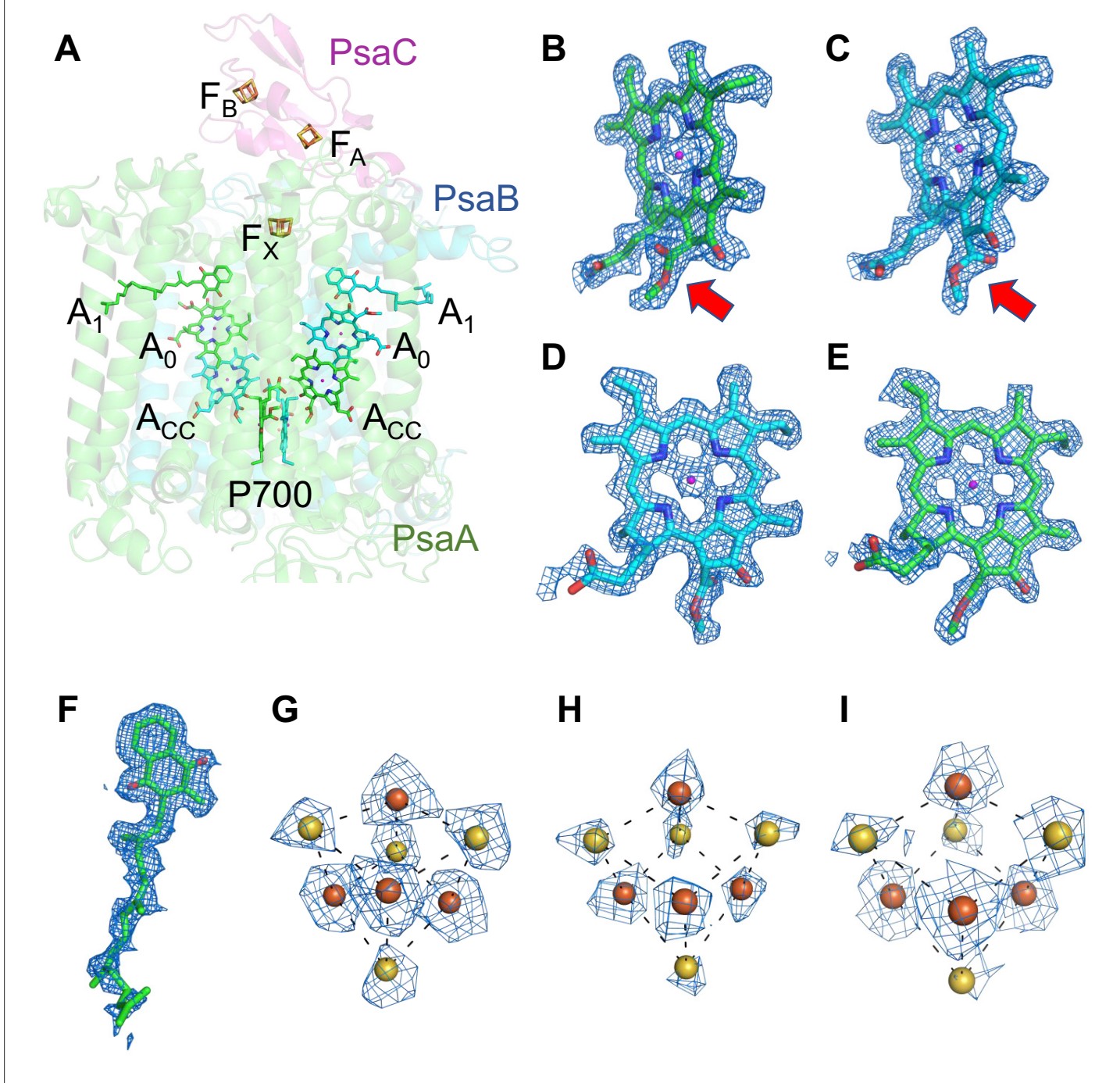

**Figure 2.** Cofactors involved in the electron-transfer reaction of the *Gloeobacter* photosystem I (PSI). (**A**) Arrangement of cofactors involved in the electron-transfer reaction. P700, special pair Chls; Acc, accessory Chl; $A_0$, primary electron acceptor; $A_1$, secondary electron acceptor menaquinone-4; $F_X$, $F_A$, and $F_B$, iron–sulfur clusters. P700 serves as a primary electron donor, and the electron produced is transferred to $A_1$ via $A_0$ (and $A_{CC}$) in each branch, followed by sequential transfer to $F_X \rightarrow F_A \rightarrow F_B$. However, whether the electrons are transferred through both branches or proceed preferentially through one branch is still under debate. (**B–I**) The cryo-electron microscopy (cryo-EM) density maps of cofactors and their refined models. (**B**) Chl *a'* in P700; (**C**) Chl *a* in P700; (**D**) Acc; (**E**) $A_0$; (**F**) $A_1$; (**G–I**), $F_X$, $F_A$, and $F_B$. Red arrows indicate structural differences between Chl *a* and Chl *a'* (**B, C**). The densities for Chls, quinone, and iron–sulfur clusters were depicted at 5 σ, 3 σ, and 15 σ, respectively.

in the green-lineage PSI cores, PsaK and PsaX are found in other cyanobacterial PSI structures (Suga and Shen, 2020; *Jordan et al., 2001*; *Malavath et al., 2018*; *Kato et al., 2020*; *Gisriel et al., 2020*; *Hamaguchi et al., 2021*; *Xu et al., 2021*). The absence of these subunits in the *Gloeobacter* PSI suggests that the PSI core from *Gloeobacter* is in an early stage of evolution before PsaK and PsaX are incorporated into PSI, and is consistent with the primordial nature of this cyanobacterium during the evolutionary process.

The root mean square deviations (RMSDs) between the monomer unit of *Gloeobacter* PSI and that of cyanobacterial PSI with the same set of subunits from *T. elongatus* (PDB: 1JB0) (*Jordan et al., 2001*), *Synechocystis* (PDB: 5OY0) (*Malavath et al., 2018*), *Thermosynechococcus vulcanus* (PDB: 6K33) (*Akita et al., 2020*), *Halomicronema hongdechloris* (PDB: 6KMW) (*Kato et al., 2020*), and *Acaryochloris marina* MBIC11017 (PDB: 7COY) (*Hamaguchi et al., 2021*), are 0.85 Å for 2,008 Cα atoms, 0.82 Å for 2,019 Cα atoms, 0.90 Å for 2,011 Cα atoms, 0.88 Å for 1,785 Cα atoms, and 1.10 Å for 1,896 Cα atoms, respectively. This suggests that the overall structures of PSI are largely similar among the cyanobacteria, if we do not include some subunits that are absent or different between different cyanobacteria. However, some regions of the structure of *Gloeobacter* PSI display significant differences compared with other cyanobacterial PSI (*Figure 3* and *Appendix 1—figures 4–6*). The *Gloeobacter* PSI possesses four types of characteristic loop structures which are named Loop1 (Tyr515–Gln529) and Loop2 (Asn652–Ser665) in PsaA (*Appendix 1—figure 4*), Loop3 (Pro717–Ile727) in PsaB (*Appendix 1—figure 5*), and Loop4 (Gln31–Asp36) in PsaF (*Appendix 1—figure 6*). These four types of loops do not exist in other cyanobacterial PSI trimers (*Jordan et al., 2001*; *Malavath et al., 2018*; *Kato et al., 2020*; *Gisriel et al., 2020*; *Hamaguchi et al., 2021*; *Xu et al., 2021*). In contrast, the motif of Pro237–Gln248 in the *Synechocystis* PsaB is absent in the *Gloeobacter* PSI (*Appendix 1—figure 5*). All these insertions and deletions are located at the periplasmic side of the PSI monomer (*Figure 3A*).

Both Loop1 of PsaA and Loop3 of PsaB do not interact with other subunits, whereas Loop2 of PsaA and Loop4 of PsaF are elongated so that both interact with PsaB. In Loop2, Leu656 of PsaA is hydrogen bonded with Pro619 of PsaB at a distance of 3.1 Å, and Val655 of PsaA is in hydrophobic interactions with PsaB-Pro619 (*Figure 3B*). In Loop4, Gln31, Gln33, and Lys35 of PsaF interact with Lys455, Glu458, Ala485, Asn486, and Asn487 of PsaB at distances of 2.5–3.1 Å (*Figure 3C*). Since our *Gloeobacter* PSI structure is solved at a high resolution, these structural differences cannot be ascribed to uncertainties due to lower resolutions, and the structural features of *Gloeobacter* PSI appear to contribute to the stability and assembly of the PSI complex in the membranes lacking thylakoids.

## Cofactors of the PSI monomer

The cofactors identified in the monomer unit of *Gloeobacter* PSI are summarized in *Appendix 1—table 2*. There are 89 Chls *a*, 20 *β*-carotenes, 3 [4Fe–4S] clusters, 2 menaquinone-4s, and 4 lipid molecules in each monomer. The location of these molecules is similar to that in other cyanobacterial PSI structures (*Jordan et al., 2001*; *Malavath et al., 2018*; *Kato et al., 2020*; *Gisriel et al., 2020*; *Hamaguchi et al., 2021*; *Xu et al., 2021*); however, six Chls are less in the *Gloeobacter* PSI than those in the *T. vulcanus* PSI (*Akita et al., 2020*) and *Synechocystis* PSI (*Malavath et al., 2018*). These Chls include one Chl in PsaA (Chl1A), one Chl in PsaF (Chl1F), two Chls in PsaJ (Chl1J, 2J), and two Chls in PsaK (Chl1K, 2K) (*Figure 4A, B*). Both Chl1K and Chl2K do not exist in the *Gloeobacter* PSI because of the absence of PsaK in the genome and the structure. Both Chl1J and Chl2J are located in the periphery of PsaJ, whose amino acid residues cannot be assigned in the *Gloeobacter* PSI structure. The map quality around Chl1F in the *Gloeobacter* PSI is very high (*Figure 4C*), indicating that Chl1F is absent in the *Gloeobacter* PSI. Chl1A is difficult to bind to the *Gloeobacter* PSI, because the conserved His residue near Chl1A in other cyanobacteria is changed to Phe243 in the *Gloeobacter* PSI (*Appendix 1—figure 4* and *Appendix 1—figure 7A*). This causes a steric hindrance between Phe243 and Chl1A in the *Gloeobacter* PSI (*Figure 4D, Appendix 1—figure 7A*), making the Chl molecule unable to bind. In addition, it is interesting to note that both *T. vulcanus* and *H. hongdechloris* PSIs possess an extra Chl (Chl1B) in PsaB (*Kato et al., 2020*; *Akita et al., 2020*), which is absent in both *Gloeobacter* PSI and *Synechocystis* PSI (*Malavath et al., 2018*). This is due to the changes in the loop structures (*Figure 4E, Appendix 1—figure 5*), as suggested by *Toporik et al., 2020*.

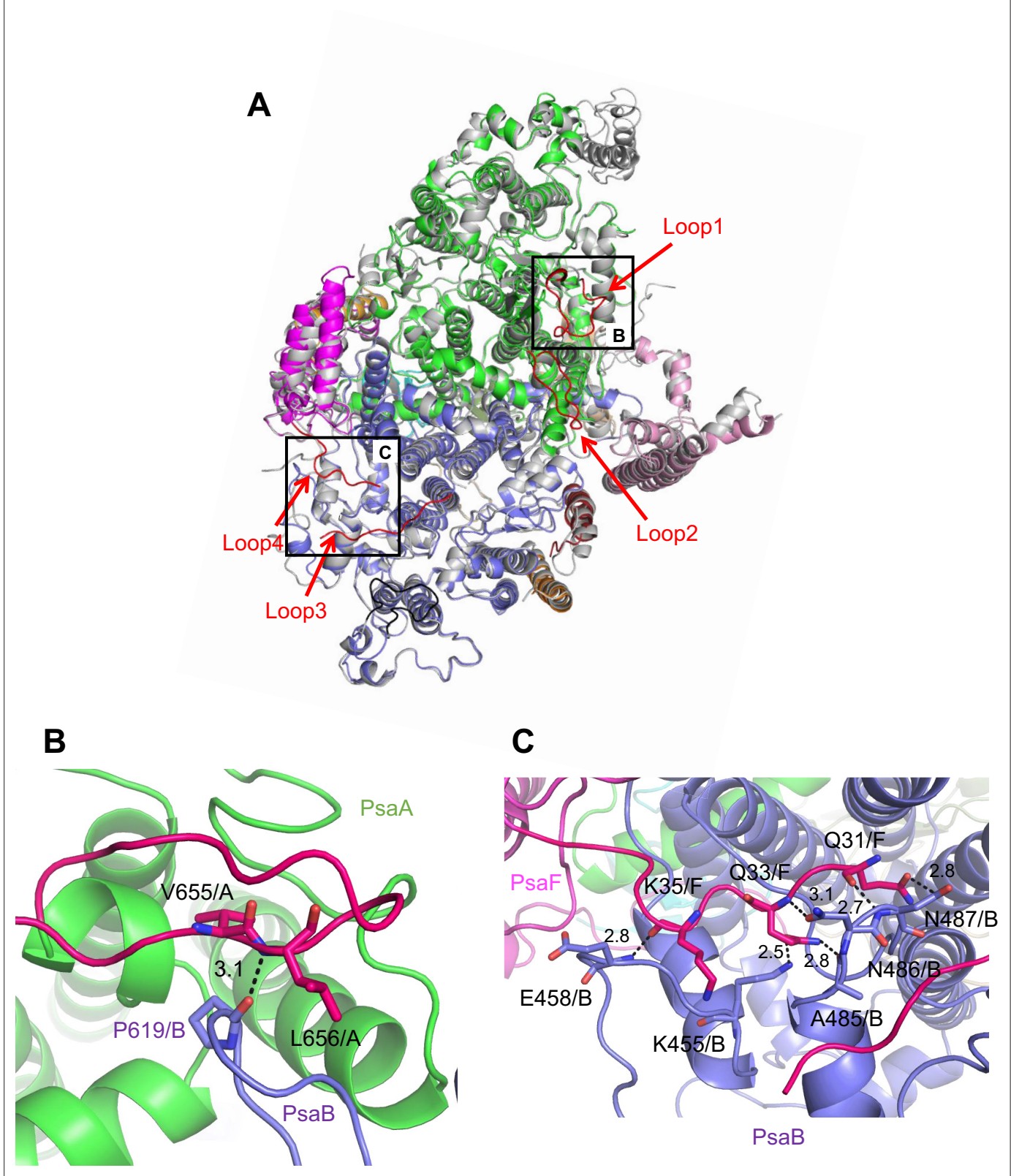

**Figure 3.** Characteristic structure of the *Gloeobacter* photosystem I (PSI). (**A**) Superposition of the structure of *Gloeobacter* PSI (colored) with that of the *Synechocystis* PSI (gray) (PDB: 5OY0) viewed along the membrane normal from the periplasmic side. Structural differences of loop insertions are colored in red. The RMSD is 0.82 Å for 2,019 Cα atoms from subunits observed commonly between *Gloeobacter* and *Synechocystis* PSIs. (**B, C**) Closeup views of the loop insertions in the regions of Asn652–Ser665 of PsaA (Loop2) in the *Gloeobacter* PSI (**B**) and of Gln31–Asp36 of PsaF (Loop4) in the *Gloeobacter* PSI (**C**).

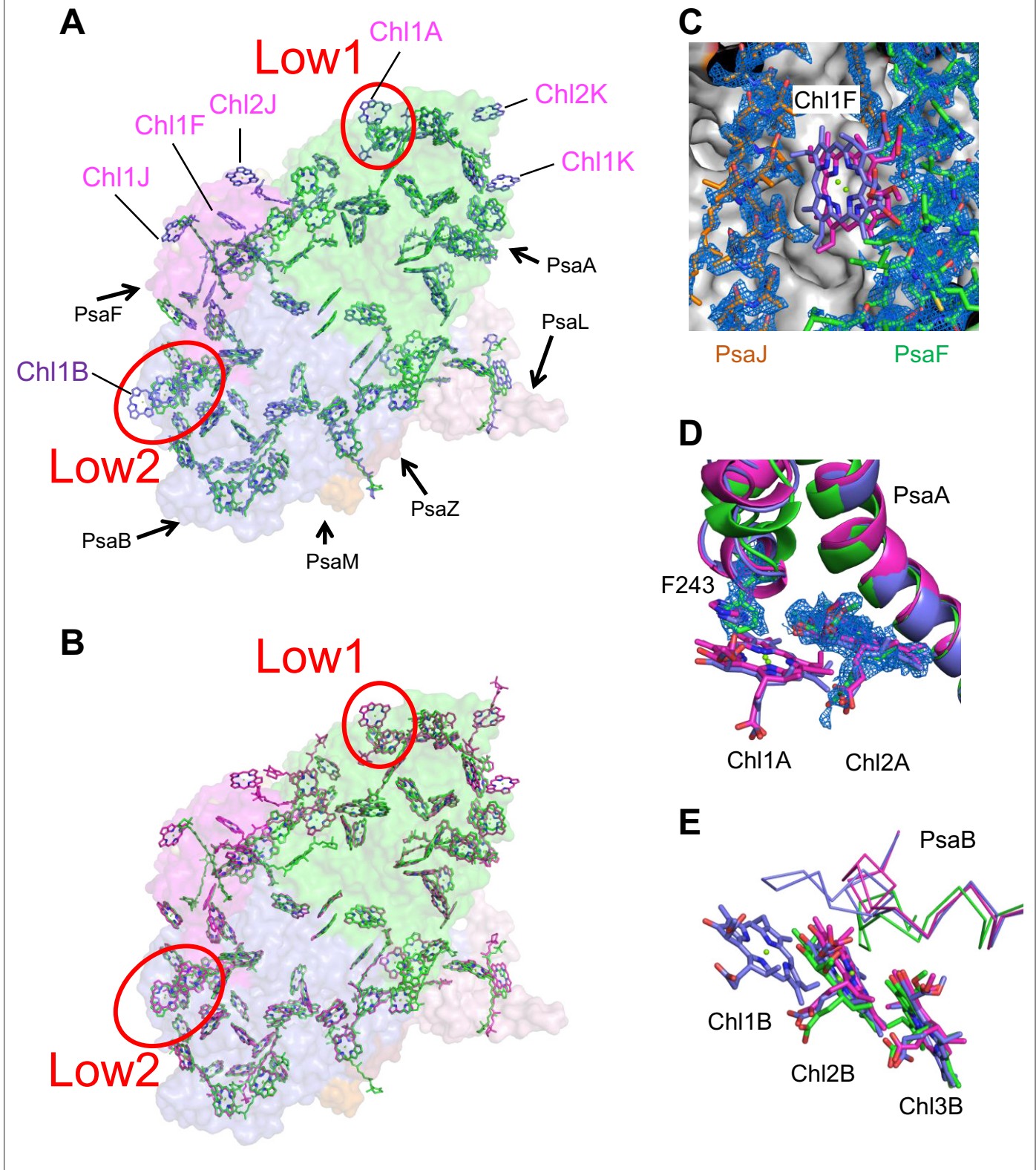

**Figure 4.** Comparison of pigments among the three types of photosystem I (PSIs). (**A, B**) Superposition of the PSI-monomer structures viewed along the membrane normal from the periplasmic side, with protein subunits depicted in a surface model. (**A**) *Gloeobacter* (green) vs. *T. vulcanus* (purple) (PDB: 6K33); (**B**) *Gloeobacter* (green) vs. *Synechocystis* (magenta) (PDB: 5OY0). Red circles stand for the sites of Low1 and Low2. Closeup views of the environments near pigments of Chl1F (**C**) and Chl1A (**D**) with their densities depicted at 1.5 σ, and Chl1B (**E**), in the three species.

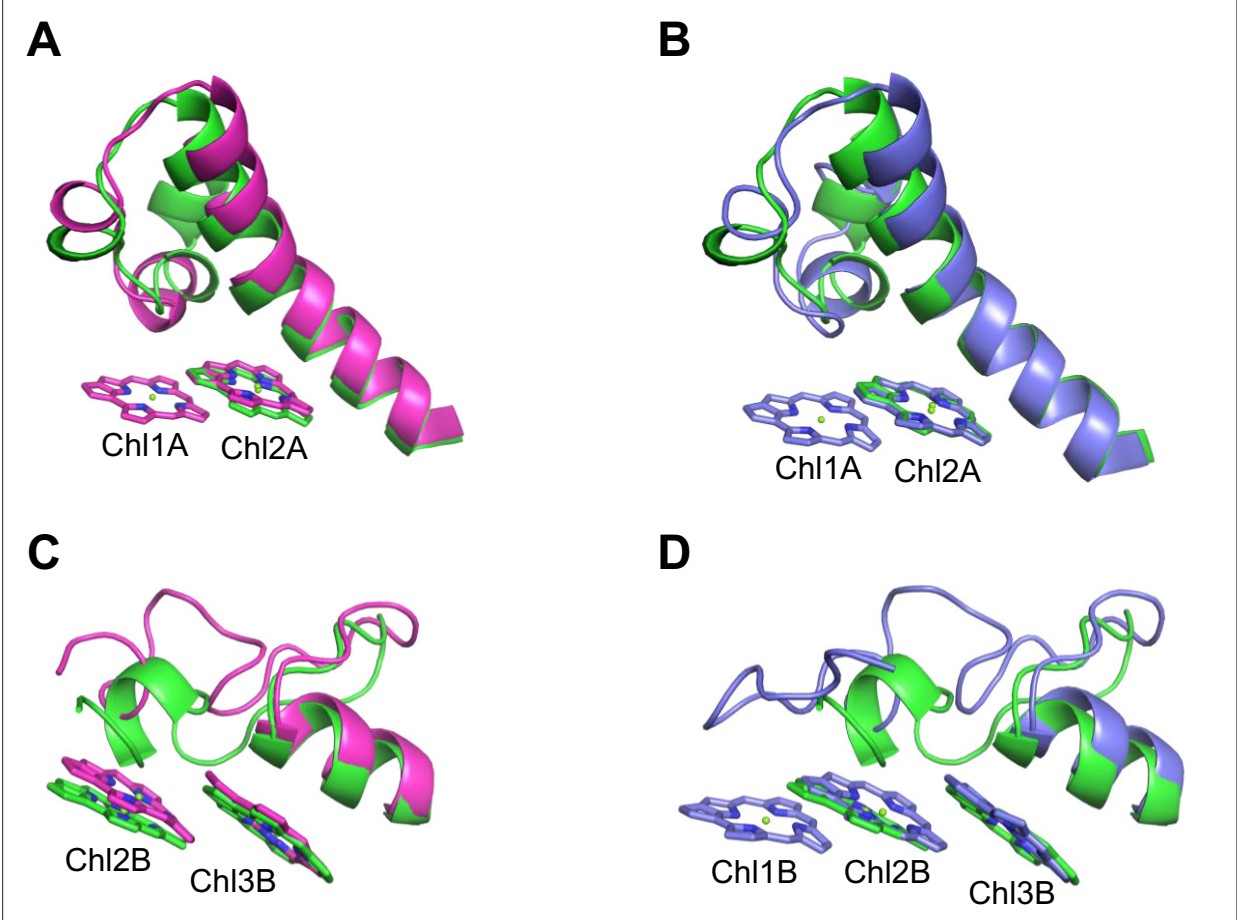

**Figure 5.** Closeup views of Low1 and Low2. (**A**) Superposition of the Low1 site in the *Gloeobacter* photosystem I (PSI; green) and *Synechocystis* PSI (magenta) (PDB: 5OY0). (**B**) Superposition of the Low1 site in the *Gloeobacter* PSI (green) and *T. vulcanus* PSI (purple) (PDB: 6K33). (**C**) Superposition of the Low2 site in the *Gloeobacter* PSI (green) and *Synechocystis* PSI (magenta). (**D**) Superposition of the Low2 site in the *Gloeobacter* PSI (green) and *T. vulcanus* PSI (purple).

## Identification of Chl clusters absent in the *Gloeobacter* PSI

The energy levels of Chls tend to be lowered by the formation of dimeric or trimeric Chls; therefore, we focused on dimeric or trimeric Chl clusters through the structural comparisons. Among the six Chl molecules that are absent in the *Gloeobacter* PSI mentioned above, Chl1A forms a dimer with Chl2A in both *Synechocystis* PSI (*Figure 5A*) and *T. vulcanus* PSI (*Figure 5B*). Thus, this Chl dimer may represent one of the low-energy Chls in most cyanobacteria other than *Gloeobacter*. The Mg atom of Chl1A is coordinated by a water molecule in the PSI structures of *T. elongatus* (*Jordan et al., 2001*) and *H. hongdechloris* (*Kato et al., 2020*). In *T. elongatus*, this water molecule is hydrogen bonded to a neighboring His240 (*Appendix 1—figure 7A*), indicating that this hydrogen bond contributes to the stability of Chl1A. Sequence alignment showed that this His240 is conserved among most cyanobacteria (*Appendix 1—figure 4*); however, it is changed to Phe243 in the *Gloeobacter* PsaA (*Appendix 1—figure 4* and *Appendix 1—figure 7A*). As mentioned above, this change leads to the loss of the water molecule that can coordinate the Chl molecule (*Appendix 1—figure 7A*). Thus, both the His residue and water molecule are responsible for the binding of Chl1A. Here, we name the Chl dimer of Chl1A and Chl2A found in the structure of most cyanobacterial PSI as Low1 (*Figure 4A, B*).

Further structural comparisons of *T. vulcanus* PSI with *Gloeobacter* PSI and *Synechocystis* PSI show a trimeric Chl cluster of Chl1B/Chl2B/Chl3B present in the *T. vulcanus* PSI but absent in the *Gloeobacter* and *Synechocystis* PSIs. The difference is due to the absence of Chl1B in the latter two cyanobacteria (*Figure 5C, D*). Therefore, this Chl trimer is also a probable candidate for low-energy Chls, which is supported by the spectroscopic analyses of PSI trimers (*Schlodder et al., 2007*) and the

cryo-EM structure of PSI mutant complexes (*Toporik et al., 2020*). Based on these findings, we name the Chl trimer of Chl1B/Chl2B/Chl3B as Low2 (*Figure 4A, B*).

## Correlation of Low1 and Low2 with characteristic fluorescence bands

Our interest is how Low1 and Low2 identified here are related to Chls fluorescing at around 723 and 730 nm observed in the steady-state fluorescence-emission spectra (*Appendix 1—figure 8A*; *van der Lee et al., 1993*; *Mimuro et al., 2010*; *Turconi et al., 1996*; *Pålsson et al., 1996*). The spectrum of *Synechocystis* PSI (red line in *Appendix 1—figure 8A*) exhibits a fluorescence peak at around 723 nm, whereas that of *T. vulcanus* PSI (blue line in *Appendix 1—figure 8A*) displays a fluorescence peak at around 731 nm. In addition, the width of the fluorescence peak from the *T. vulcanus* PSI is broader than that from the *Synechocystis* PSI; therefore, it seems highly probable that the *T. vulcanus* PSI also possesses Chls fluorescing at around 723 nm in addition to those at around 730 nm. In contrast, the fluorescence spectrum of *Gloeobacter* PSI (black line in *Appendix 1—figure 8A*) exhibits a maximum peak at around 694 nm, which is significantly defferent from the fluorescence peaks at around 723 and 730 nm. The characteristic differences of fluorescence peaks in these cyanobacteria can even be observed in the absorption spectra measured at 77 K, which exhibit different intensities of relative absorbance over 700 nm (*Appendix 1—figure 8B*).

Based on these structural and fluorescence properties, it is proposed that Low1 and Low2 are Chls fluorescing at around 723 and 730 nm, respectively. In a previous study, *Schlodder et al., 2007* listed nine groups of Chls as potential candidates for low-energy Chls in PSI based on their spectroscopic studies. Eight of the nine groups of Chls are well conserved in the PSI structures among the three cyanobacteria (*Appendix 1—figure 9*). Therefore, it is highly possible that these eight Chl groups are not Chls fluorescing at around 723 or 730 nm. In contrast, the remaining Chl group (*Schlodder et al., 2007*) is a triply stacked Chls, B1231/B1232/B1233 (the Chl numbering taken from *Jordan et al., 2001*), which is consistent with Low2 presented here. These observations support that the Chl cluster of Chl1B/Chl2B/Chl3B is Low2 which is present in *T. vulcanus* and *T. elongatus* but absent in *Synechocystis* and *Gloeobacter*.

In the previous study, the Low1 site was not listed as potential low-energy Chls (*Schlodder et al., 2007*). This may be due to the large number of Chls in the PSI core, resulting in difficulties in identifying individual low-energy Chls using spectroscopic methods. Our structure clearly indicates that the Low1 site is absent in the *Gloeobacter* PSI but present in other cyanobacteria, leading to the idea that Low1 is also a low-energy Chl site responsible for the long-fluorescence emission observed in other cyanobacteria but not in *Gloeobacter*.

Some cyanobacteria have tetrameric PSI complexes whose structures have been solved from the cyanobacterium *Anabaena* sp. PCC 7120 (hereafter referred to as *Anabaena*) (*Kato et al., 2019*; *Zheng et al., 2019*; *Chen et al., 2020*). Both Low1 and Low2 exist in the monomer units of the *Anabaena* PSI tetramer (*Appendix 1—figure 7B, C*). The fluorescence spectrum of *Anabaena* PSI tetramer showed a peak at around 730 nm under the liquid-nitrogen condition (*Kato et al., 2019*; *Nagao et al., 2020b*), which is similar to the spectrum of the *T. vulcanus* PSI trimer. These observations are in line with our proposal that Low1 and Low2 are responsible for Chls fluorescing at around 723 and 730 nm, respectively, in most cyanobacterial PSI monomers other than *Gloeobacter*, irrespective of the oligomeric states of PSI. However, it should be noted that the energy levels of Chls cannot be assigned only by the structural analysis of PSI. Further mutagenesis studies and theoretical calculations will be required for understanding the correlation of Low1 and Low2 with the fluorescence peaks at around 723 and 730 nm.

## Low-energy Chls in interfaces among PSI-monomer units

Low-energy Chls including Chls fluorescing at around 730 nm have been thought to be located in interfaces among monomers in a PSI trimer from *T. elongatus* (*El-Mohsnawy et al., 2010*; *Çoruh et al., 2021*). The PSI monomer isolated from *T. elongatus* exhibited a fluorescence peak shifted to about 720 nm compared with a fluorescence peak at around 730 nm in the PSI trimer (*Çoruh et al., 2021*). However, the structure of PSI monomer displayed the lack of a Chl of B1233 at the Low2 site, in addition to several peripheral Chls (*Çoruh et al., 2021*). Thus, the fluorescence-peak shift observed in the *T. elongatus* PSI monomer may be due to the absence of Low2 but not Chls located in interfaces around the PSI-monomer units.

As for other cyanobacteria, *Anabaena* has both Low1 and Low2, while the fluorescence peak of PSI monomer at around 730 nm was virtually identical to that of PSI tetramer (*Kato et al., 2019*; *Nagao et al., 2020b*). Moreover, *Synechocystis* has Low1 without Low2, while the fluorescence peak of PSI monomer was consistent with that of PSI trimer (*Turconi et al., 1996*). These observations support the idea that Low1 and Low2 are responsible for Chls fluorescing at around 723 and 730 nm, respectively, and that these long-wavelength fluorescing Chls are not identical to Chls found in the interfaces among monomers, irrespective of the oligomeric states of PSI.

## Evolution of Low1 and Low2 in oxyphototrophs

The overall structures of PSI supercomplexes in complex with light-harvesting complexes I have been solved from a red alga (*Pi et al., 2018*; *Antoshvili et al., 2019*), green algae (*Su et al., 2019*; *Suga et al., 2019*; *Qin et al., 2019*; *Perez-Boerema et al., 2020*), a diatom (*Nagao et al., 2020a*; *Xu et al., 2020*), a moss (*Yan et al., 2021*), and a higher plant (*Qin et al., 2015*; *Mazor et al., 2017*). Compared with the structure of *Gloeobacter* PSI (*Figure 6A*), Low1 is present in the PSI cores from a red alga *Cyanidioschyzon merolae* (PDB: 5ZGB), a green alga *Chlamydomonas reinhardtii* (PDB: 6JO5), a diatom *Chaetoceros gracilis* (PDB: 6L4U), a moss *Physcomitrella patens* (PDB: 6L35), and a higher plant *Pisum sativum* (PDB: 4XK8). The His residue corresponding to PsaA-His240 in *T. elongatus* is also conserved among the eukaryotes (*Figure 6B*), reflecting the presence of Low1 in eukaryotes.

Low2 exists in PSIs of *P. patens* and *P. sativum* in addition to *T. vulcanus* (*Figure 7A*); however, the triple stacking interactions among these three Chls, especially the orientation of Chl1B and Chl1G, differ between cyanobacteria and the eukaryotes (*Figure 7B*). This is likely due to the different loop structures interacting with Chl1B between *T. vulcanus* and the eukaryotes (*Figure 7B, C*). In the *P. sativum* PSI, Chl1G corresponding to Chl1B is bound to Tyr93 of PsaG via a water molecule (*Appendix 1—figure 10A*). PsaG is absent in *Gloeobacter* but present in *C. reinhardtii*; however, the characteristic Tyr residue is changed to Gly in *C. reinhardtii* (*Appendix 1—figure 10B*), leading to the loss of this water molecule in *C. reinhardtii*. This suggests that the hydrogen-bond interaction between the water molecule and Tyr93 of PsaG is required for the formation of Low2 in eukaryotes. In contrast, the gene of *psaG* is not found in the red alga and diatom, resulting in the absence of Low2 in these organisms.

## Conclusions

This study reveals the high-resolution structure of the PSI trimer from *Gloeobacter* at 2.04 Å by cryo-EM. Structural comparisons of the *Gloeobacter* PSI with the *Synechocystis* and *T. vulcanus* PSIs provide evidence for the locations of two types of characteristic Chl clusters, Low1 and Low2, present in other cyanobacteria but absent in *Gloeobacter*. Based on the structural and fluorescence properties, Low1 and Low2 may be responsible for Chls fluorescing at around 723 and 730 nm, respectively, in most cyanobacteria. In particular, Low1 is conserved in most oxyphototrophs from prokaryotes to eukaryotes other than *Gloeobacter*, implying that the lack of Low1 is a characteristic of primordial cyanobacteria occurring near the origin of oxyphototrophs.

## Materials and methods
### Purification and characterization of the PSI trimer from *Gloeobacter*

The cyanobacterium *G. violaceus* PCC 7421 was grown in BG11 medium (*Stanier et al., 1971*) supplemented with 10 mM 2-[4-(2-Hydroxyethyl)-1-piperazinyl]ethanesulfonic acid (HEPES)–KOH (pH 8.0) and 1/1000 volume of KW21 (Daiichi Seimo) at a photosynthetic photon flux density (PPFD) of 5 μmol photons $m^{-2}\,s^{-1}$ at 20°C with bubbling of air containing 3% (vol/vol) $CO_2$. KW21 is helpful for enhancing the growth of photosynthetic organisms (*Nagao et al., 2007*; *Nagao et al., 2013a*). Cytoplasmic membranes were prepared after disruption of the cells with glass beads with a method similar to the preparation of thylakoid membranes as described previously (*Nagao et al., 2017*), and suspended in a buffer containing 0.2 M trehalose, 20 mM 2-morpholinoethanesulfonic acid (MES)–NaOH (pH 6.5), 5 mM $CaCl_2$, and 10 mM $MgCl_2$ (buffer A). The membranes were solubilized with 1% (wt/vol) *n*-dodecyl-$\beta$-D-maltoside ($\beta$-DDM) at a Chl concentration of 0.25 mg $ml^{-1}$ for 30 min on ice in the dark with gentle stirring. After centrifugation at 50,000 × *g* for 30 min at 4°C, the resultant supernatant was loaded onto a Q-Sepharose anion-exchange column (2.5 cm of inner diameter and 6 cm of length)

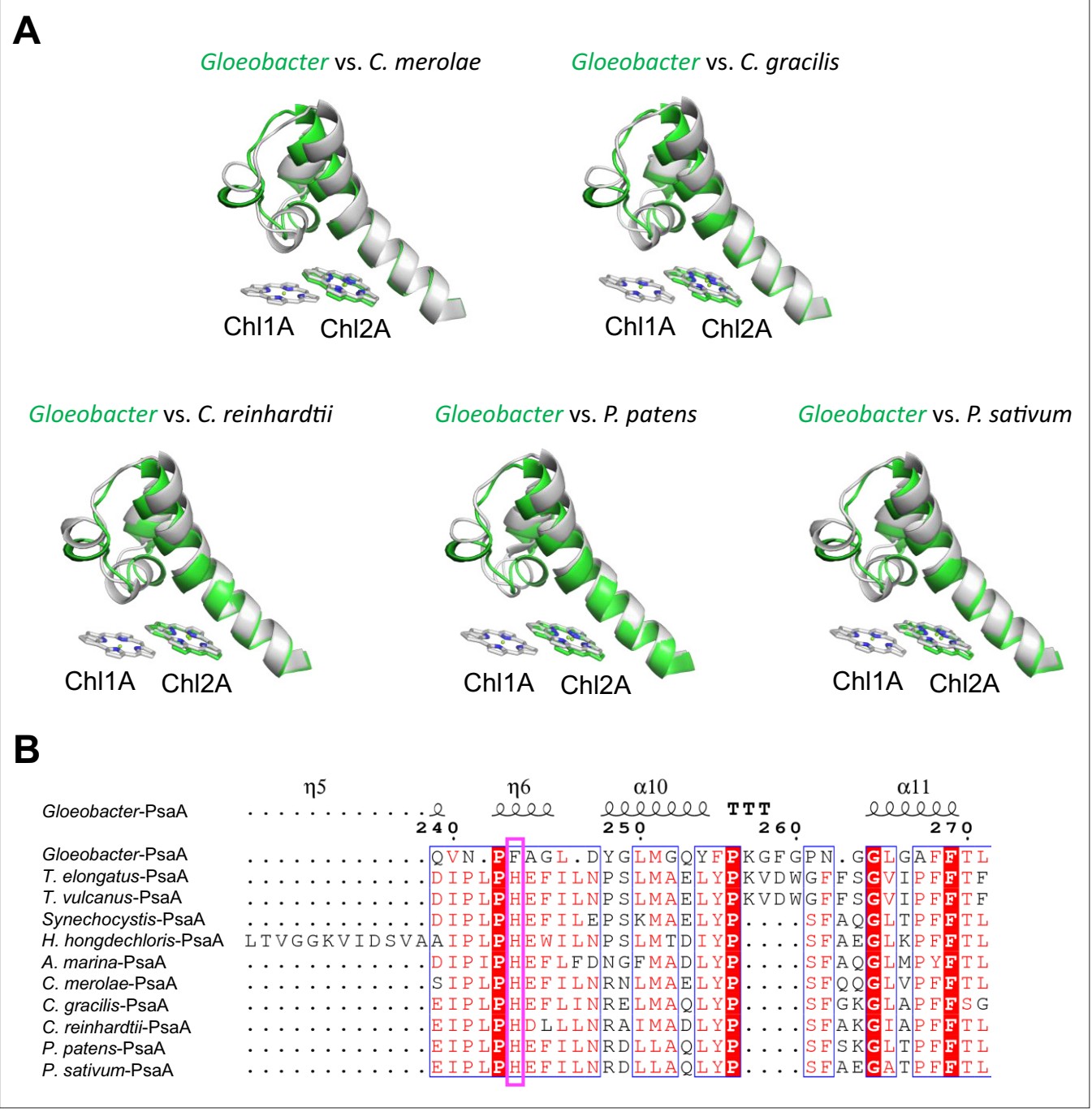

**Figure 6.** Structural comparisons of the Low1 site among oxyphototrophs. (**A**) Superposition of the Low1 site of the *Gloeobacter* photosystem I (PSI; green) with PSI complexes of photosynthetic eukaryotes (gray): *Cyanidioschyzon merolae* (PDB: 5ZGB), *Chaetoceros gracilis* (PDB: 6L4U), *Chlamydomonas reinhardtii* (PDB: 6JO5), *Physcomitrella patens* (PDB: 6L35), and *Pisum sativum* (PDB: 4XK8). (**B**) Multiple sequence alignment of PsaA among oxyphototrophs using ClustalW and ESPript. The species shown are *Gloeobacter violaceus* PCC 7421, *Thermosynechococcus elongatus* BP-1, *Thermosynechococcus vulcanus* NIES-2134, *Synechocystis* sp. PCC 6803, *Halomicronema hongdechloris* C2206, *Acaryochloris marina* MBIC11017, *C. merolae*, *C. gracilis*, *C. reinhardtii*, *P. patens*, and *P. sativum*. The pink box displays the histidine residue involved in the binding of Chl1A.

equilibrated with buffer A containing 0.03% $\beta$-DDM (buffer B). The column was washed with buffer B containing 100 mM NaCl (buffer C) until the eluate became colorless, and further washed with 60 ml of buffer B containing 150 mM NaCl. The PSI fraction was eluted with buffer B containing 200 mM NaCl, and subsequently loaded onto a linear trehalose gradient of 10–40% (wt/vol) in a medium containing 20 mM MES–NaOH (pH 6.5), 5 mM CaCl₂, 10 mM MgCl₂, 100 mM NaCl, and 0.03% $\beta$-DDM. After

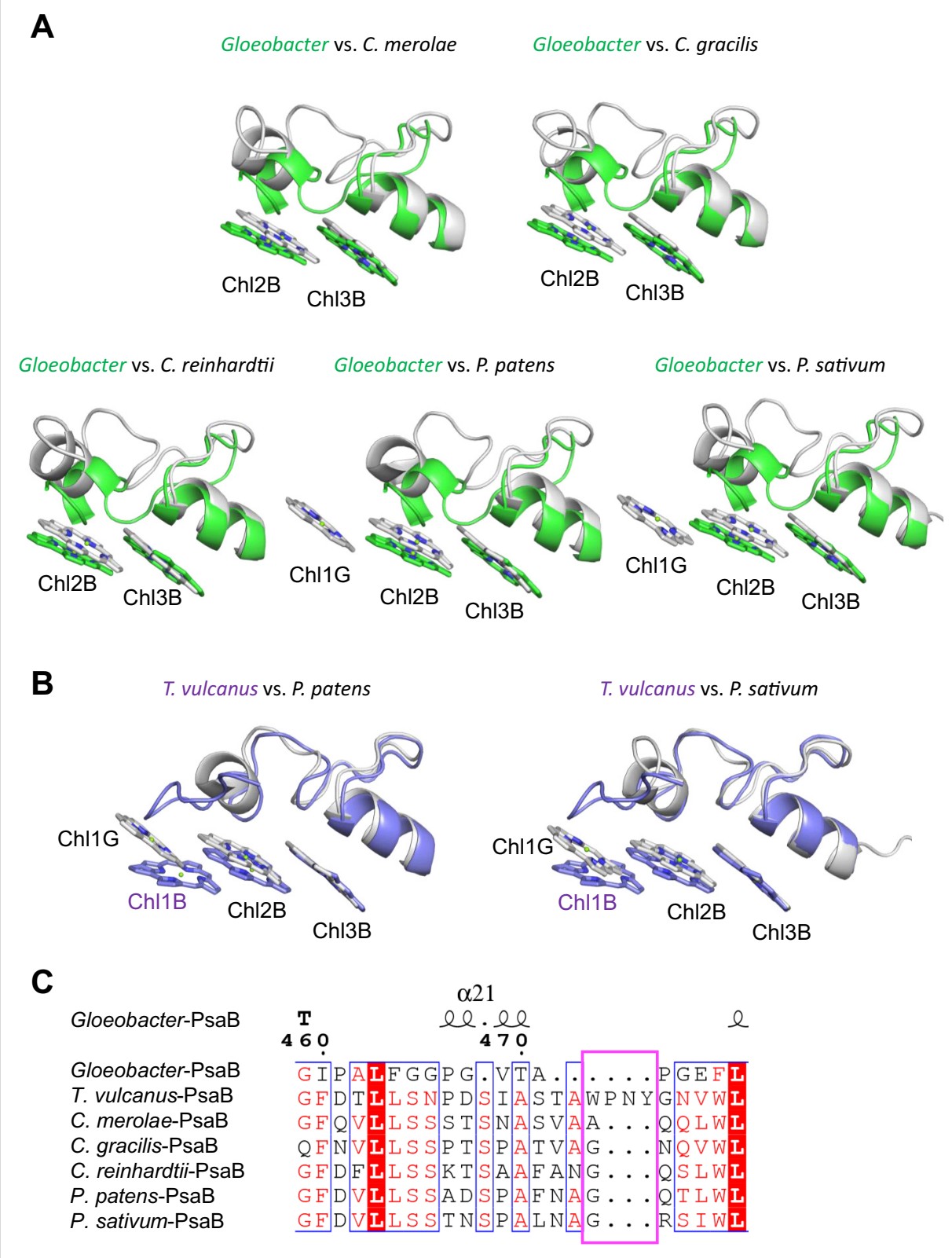

**Figure 7.** Structural comparisons of the Low2 site among oxyphototrophs. (**A**) Superposition of the Low2 site of the *Gloeobacter* photosystem I (PSI; green) with PSI complexes of photosynthetic eukaryotes (gray): *Cyanidioschyzon merolae* (PDB: 5ZGB), *Chaetoceros gracilis* (PDB: 6L4U), *Chlamydomonas reinhardtii* (PDB: 6JO5), *Physcomitrella patens* (PDB: 6L35), and *Pisum sativum* (PDB: 4XK8). (**B**) Superposition of the Low2 site of *T. vulcanus* PSI (purple) (PDB: 6K33) with PSI complexes (gray) of *P. patens* and *P. sativum*. (**C**) Multiple sequence alignment of PsaB using ClustalW and ESPript. The pink box indicates the loop involved in the binding of Chl1B in *T. vulcanus*.

centrifugation at 154,000 × *g* for 18 hr at 4°C (P40ST rotor; Hitachi), a band containing the PSI trimers was collected and then concentrated using a 150 kDa cutoff filter (Apollo; Orbital Biosciences) at 4,000 × *g*. The concentrated PSI-trimer complexes were stored in liquid nitrogen until use.

Subunit composition of the PSI trimer was analyzed by sodium dodecyl sulfate–polyacrylamide gel electrophoresis containing 16% acrylamide and 7.5 M urea according to a method (*Ikeuchi and Inoue, 1988*; *Appendix 1—figure 1A*, *Source data 1*). The samples (5 µg of Chl) were solubilized by 3% lithium lauryl sulfate and 75 mM dithiothreitol for 10 min at 60°C, and loaded onto the gel. A standard molecular weight marker (SP-0110; APRO Science) was used. The subunit bands were assigned by mass spectrometry according to a previous method (*Nagao et al., 2019*). Pigment compositions were analyzed as described in *Nagao et al., 2013b*, and the elution profile was monitored at 440 nm (*Appendix 1—figure 1B*).

## Purification of the PSI trimer from *Synechocystis*

The cyanobacterium *Synechocystis* sp. PCC 6803 47H strain was grown in the BG11 medium supplemented with 10 mM HEPES–KOH (pH 8.0) at the PPFD of 30 µmol photons m$^{-2}$ s$^{-1}$ at 30°C with bubbling of air containing 3% (vol/vol) CO$_2$. The 47H strain has a six-histidine tag at the C-terminus of the CP47 subunit (*Ito and Tanaka, 2011*). Thylakoid membranes were prepared as described previously (*Nagao et al., 2017*) and suspended in buffer A. The thylakoids were solubilized with 1% *β*-DDM at a Chl concentration of 0.50 mg ml$^{-1}$ for 30 min on ice in the dark with gentle stirring. After centrifugation at 50,000 × *g* for 10 min at 4°C, the resultant supernatant was loaded onto a Ni$^{2+}$ affinity column (2.5 cm of inner diameter and 10 cm of length) equilibrated with buffer C. The PSI-enriched fraction was collected by washing the column with buffer C, and subsequently diluted with an equal volume of buffer B. The diluted sample was applied onto a Q-Sepharose anion-exchange column (2.5 cm of inner diameter and 10 cm of length) equilibrated with buffer B. The column was washed with buffer B containing 150 mM NaCl until the eluate became colorless, and further washed with 50 ml of buffer B containing 200 mM NaCl and subsequently with 100 ml of buffer B containing 250 mM NaCl. The PSI fraction was eluted with buffer B containing 300 mM NaCl, and subsequently loaded onto a linear trehalose gradient of 10–40% (wt/vol) in a medium containing 20 mM MES–NaOH (pH 6.5), 5 mM CaCl$_2$, 10 mM MgCl$_2$, 100 mM NaCl, and 0.03% *β*-DDM. After centrifugation at 154,000 × *g* for 18 hr at 4°C (P40ST rotor; Hitachi), a band containing the PSI trimers was collected and then concentrated using a 150 kDa cutoff filter (Apollo; Orbital Biosciences) at 4,000 × *g*. The concentrated PSI-trimer complexes were stored in liquid nitrogen until use.

## Purification of the PSI trimer from *T. vulcanus*

The cyanobacterium *T. vulcanus* NIES-2134 was grown and its thylakoid membranes were prepared as described previously (*Kawakami and Shen, 2018*). The PSI-enriched fraction was obtained as precipitation by centrifugation at 100,000 × *g* for 60 min at 4°C after second-round treatments of the thylakoids with *N,N*-dimethyldodecylamine-*N*-oxide (*Kawakami and Shen, 2018*), and suspended with a 30 mM MES–NaOH (pH 6.0) buffer containing 5% (wt/vol) glycerol, 3 mM CaCl$_2$, and 0.03% *β*-DDM. The PSI fraction was loaded onto a linear trehalose gradient of 10–40% (wt/vol) in a medium containing 20 mM MES–NaOH (pH 6.5), 5 mM CaCl$_2$, 10 mM MgCl$_2$, 100 mM NaCl, and 0.03% *β*-DDM. After centrifugation at 154,000 × *g* for 18 hr at 4°C (P40ST rotor; Hitachi), a band containing the PSI trimers was collected and then concentrated using a 150 kDa cutoff filter (Apollo; Orbital Biosciences) at 4,000 × *g*. The concentrated PSI-trimer complexes were stored in liquid nitrogen until use.

## Absorption and fluorescence-emission spectra at 77 K

Absorption spectra of the three types of PSI trimers were measured at 77 K using a spectrometer equipped with an integrating sphere unit (V-650/ISVC-747, JASCO) (*Hamada et al., 2012*), and their steady-state fluorescence-emission spectra were recorded at 77 K using a spectrofluorometer (FP-8300/PMU-183, JASCO). The excitation wavelength was set to 440 nm.

## Cryo-EM data collection

For cryo-EM experiments, 3 µl aliquots of the *Gloeobacter* PSI trimers (1.68 mg Chl ml$^{-1}$) were applied to Quantifoil R1.2/1.3, Cu 200 mesh grids pretreated by gold sputtering. Without waiting for incubation, the excess amount of the solution was blotted off for 5 s with a filter paper in an FEI Vitrobot

Mark IV at 4°C under 100% humidity. The grids were plunged into liquid ethane cooled by liquid nitrogen and then transferred into a CRYO ARM 300 electron microscope (JEOL) equipped with a cold-field emission gun operated at 300 kV. Zero-energy loss images were recorded at a nominal magnification of ×60,000 on a direct electron detection camera (Gatan K3, AMETEK) with a nominal defocus range of −1.8 to −0.6 μm. One-pixel size corresponded to 0.823 Å. Each image stack was exposed at a dose rate of 17.555 e⁻ Å⁻² s⁻¹ for 4.0 s, and consisted of dose-fractionated 50 movie frames. In total 7,282 image stacks were collected.

## Cryo-EM image processing

The movie frames thus obtained were aligned and summed using MotionCor2 (*Zheng et al., 2017*) to yield dose weighted images. Estimation of the contrast transfer function (CTF) was performed using CTFFIND4 (*Mindell and Grigorieff, 2003*). All of the following processes were performed using RELION3.1 (*Zivanov et al., 2020*). In total 961,960 particles were automatically picked up and used for reference-free 2D classification. Then, 946,827 particles were selected from good 2D classes and subsequently subjected to 3D classification without imposing any symmetry. An initial model for the first 3D classification was generated de novo from 2D classifications. As shown in *Appendix 1—figure 2*, the final PSI-trimer structure was reconstructed from 261,743 particles. The overall resolution of the cryo-EM map was estimated to be 2.04 Å by the gold standard FSC curve with a cutoff value of 0.143 (*Appendix 1—figure 2D*; *Grigorieff and Harrison, 2011*). Local resolutions were calculated using RELION (*Appendix 1—figure 2F*).

## Model building and refinement

The cryo-EM map thus obtained was used for the model building of the PSI trimer. Each subunit of the homology models constructed using the Phyre2 server (*Kelley et al., 2015*) was first manually fitted into the map by using UCSF Chimera (*Pettersen et al., 2004*), and then inspected and manually adjusted with Coot (*Emsley et al., 2010*). The complete PSI-trimer structure was refined with phenix.real_space_refine (*Adams et al., 2010*) and REFMAC5 (*Murshudov et al., 2011*) with geometric restraints for the protein-cofactor coordination. The final model was validated with MolProbity (*Chen et al., 2010*), EMRinger (*Barad et al., 2015*), and Q-score (*Pintilie et al., 2020*). The statistics for all data collection and structure refinement are summarized in *Appendix 1—Tables 1 and 2*. All structural figures were made by PyMOL (*Schrödinger, 2015*) and UCSF ChimeraX (*Goddard et al., 2018*).

## Acknowledgements

This work was supported by JSPS KAKENHI grant nos. JP20H02914 (Koji.K.), JP21K19085 (R.N.), JP20K06528 (Keisuke.K), JP16H06553 (S.A.), and JP17H06433 (J.-R.S.), JST-Mirai Program Grant Number JPMJMI20G5 (K.Y.), Takeda Science Foundation (Koji.K.), and the Cyclic Innovation for Clinical Empowerment (CiCLE) from the Japan Agency for Medical Research and Development, AMED (T.H., Keisuke.K., and K.Y.).

## Additional information

### Funding

| Funder | Grant reference number | Author |
|---|---|---|
| Japan Society for the Promotion of Science | JP20H02914 | Koji Kato |
| Japan Society for the Promotion of Science | JP21K19085 | Ryo Nagao |
| Japan Society for the Promotion of Science | JP20K06528 | Keisuke Kawakami |
| Japan Society for the Promotion of Science | JP16H06553 | Seiji Akimoto |

| Funder | Grant reference number | Author |
|---|---|---|
| Japan Society for the Promotion of Science | JP17H06433 | Jian-Ren Shen |
| JST-Mirai | JPMJMI20G5 | Koji Yonekura |
| Takeda Science Foundation | | Koji Kato |
| the Cyclic Innovation for Clinical Empowerment (CiCLE) from the Japan Agency for Medical Research and Development, AMED | | Tasuku Hamaguchi Keisuke Kawakami Koji Yonekura |

The funders had no role in study design, data collection, and interpretation, or the decision to submit the work for publication.

## Author contributions

Koji Kato, Formal analysis, Funding acquisition, Investigation, Validation, Writing – original draft; Tasuku Hamaguchi, Formal analysis, Funding acquisition, Investigation, Validation; Ryo Nagao, Conceptualization, Formal analysis, Funding acquisition, Investigation, Project administration, Validation, Writing – original draft, Writing – review and editing; Keisuke Kawakami, Validation; Yoshifumi Ueno, Takehiro Suzuki, Formal analysis, Investigation, Validation; Hiroko Uchida, Akio Murakami, Naoshi Dohmae, Formal analysis, Resources; Yoshiki Nakajima, Makio Yokono, Formal analysis; Seiji Akimoto, Formal analysis, Funding acquisition, Investigation, Resources, Supervision, Validation; Koji Yonekura, Funding acquisition, Project administration, Resources, Supervision, Validation; Jian-Ren Shen, Conceptualization, Funding acquisition, Project administration, Resources, Supervision, Validation, Writing – review and editing

## Author ORCIDs

Ryo Nagao http://orcid.org/0000-0001-8212-3001
Seiji Akimoto http://orcid.org/0000-0002-8951-8978
Naoshi Dohmae http://orcid.org/0000-0002-5242-9410
Koji Yonekura http://orcid.org/0000-0001-5520-4391
Jian-Ren Shen http://orcid.org/0000-0003-4471-8797

## Decision letter and Author response

Decision letter https://doi.org/10.7554/eLife.73990.sa1
Author response https://doi.org/10.7554/eLife.73990.sa2

# Additional files

## Supplementary files
• Transparent reporting form
• Source data 1. The raw data of SDS-PAGE.

## Data availability

The cryo-EM density map and atomic model have been deposited in the Electron Microscopy Data Bank and the Protein Data Bank (EMD ID code 31455 and PDB ID code 7F4V). Figs. 1-7 in the main text are made with these data by PyMOL and UCSF ChimeraX. The source data of Appendix 1—figure 1A is provided as a xlsx file, which is named Source Data 1.

The following datasets were generated:

| Author(s) | Year | Dataset title | Dataset URL | Database and Identifier |
| --- | --- | --- | --- | --- |
| Kato K, Hamaguchi T, Nagao R, Kawakami K, Yonekuru K, Shen J-R | 2022 | Cryo-EM Structure of a primordial cyanobacterial photosystem I | https://www.ebi.ac.uk/emdb/EMD-31455 | Electron Microscopy Data Bank, 31455 |
| Kato K, Hamaguchi T, Nagao R, Kawakami K, Yonekuru K, Shen J-R | 2022 | Cryo-EM Structure of a primordial cyanobacterial photosystem I | https://www.rcsb.org/structure/7F4V | RCSB Protein Data Bank, 7F4V |

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

## Appendix 1

### Supplementary Information

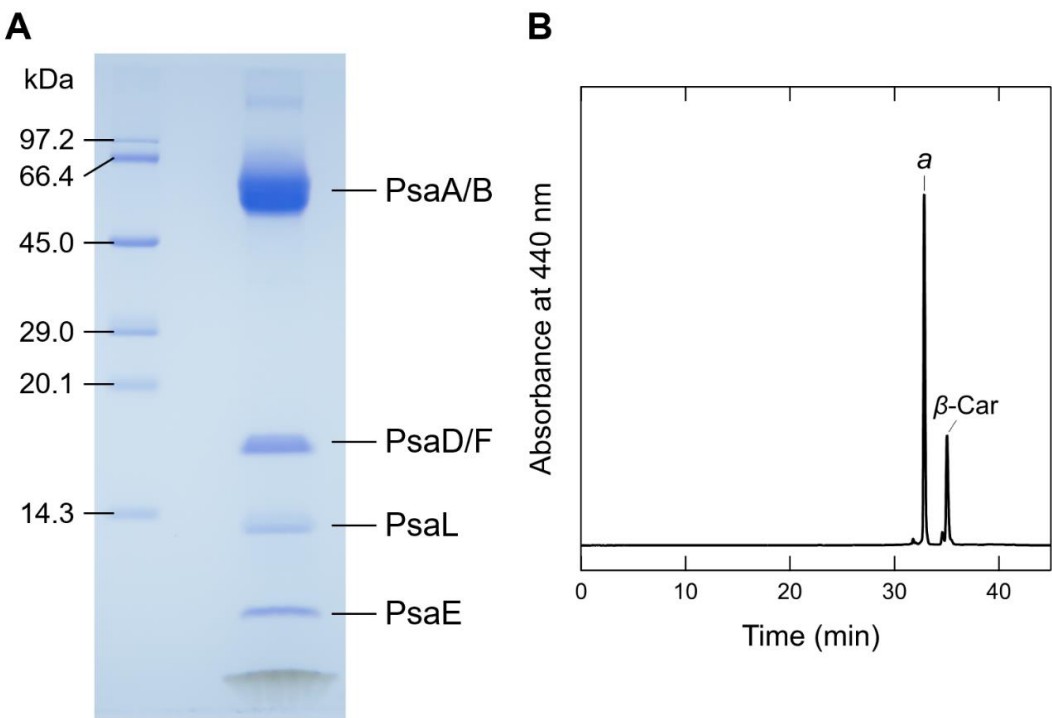

**Appendix 1—figure 1.** Biochemical characterization of the *Gloeobacter* PSI trimer. (**A**) SDS-PAGE analysis of the PSI trimer. Protein bands were identified by mass spectrometry. It should be noted that some of the subunits in the PSI trimer were not detected by SDS-PAGE, probably due to their poor staining by CBB, but were visualized by the present cryo-EM map. (**B**) HPLC analysis of pigments extracted from the PSI trimer monitored at 440 nm. Letters of *a* and *β*-Car indicate Chl *a* and *β*-carotene, respectively. Data are representative of three independent experiments.

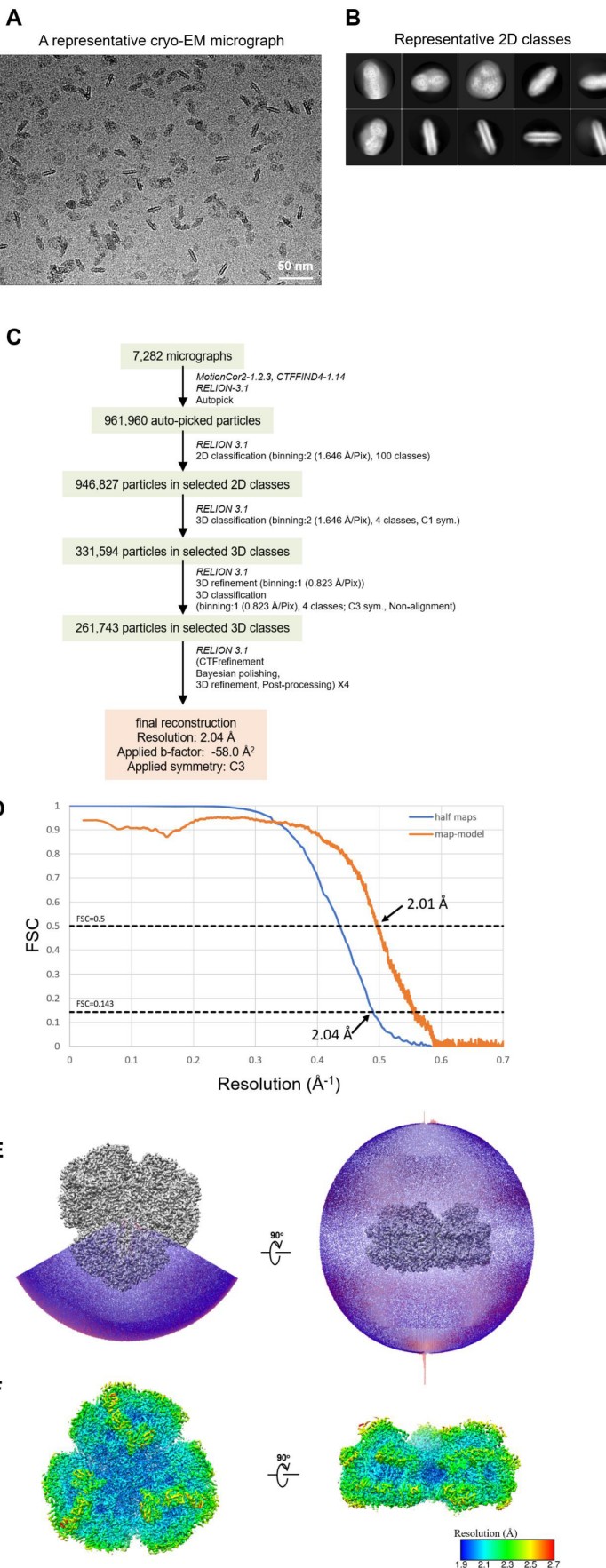

**A** A representative cryo-EM micrograph

**B** Representative 2D classes

**C**

7,282 micrographs

*MotionCor2-1.2.3, CTFFIND4-1.14*
*RELION-3.1*
Autopick

961,960 auto-picked particles

*RELION 3.1*
2D classification (binning:2 (1.646 Å/Pix), 100 classes)

946,827 particles in selected 2D classes

*RELION 3.1*
3D classification (binning:2 (1.646 Å/Pix), 4 classes, C1 sym.)

331,594 particles in selected 3D classes

*RELION 3.1*
3D refinement (binning:1 (0.823 Å/Pix))
3D classification
(binning:1 (0.823 Å/Pix), 4 classes; C3 sym., Non-alignment)

261,743 particles in selected 3D classes

*RELION 3.1*
(CTFrefinement
Bayesian polishing,
3D refinement, Post-processing) X4

final reconstruction
Resolution: 2.04 Å
Applied b-factor: -58.0 Å$^2$
Applied symmetry: C3

**D**

**E**

**F**

Resolution (Å)
1.9  2.1  2.3  2.5  2.7

**Appendix 1—figure 2.** Cryo-EM data collection and processing of the *Gloeobacter* PSI trimer. (**A**) Representative cryo-EM micrograph of the PSI trimer. (**B**) Representative 2D classes of the PSI trimer. The box size is 330 Å. (**C**) A flowchart showing the classification scheme of the PSI trimer. The overall structure of PSI was reconstructed at a 2.04-Å resolution from 261,743 particles. (**D**) FSC curves of the PSI trimer for independently refined half maps (half maps) and full map vs. model (map-model). (**E**) Angular distribution of the particles used for reconstruction of the PSI trimer. Each cylinder represents one view, and its height is proportional to the number of particles. (**F**) Local resolution map of the PSI trimer.

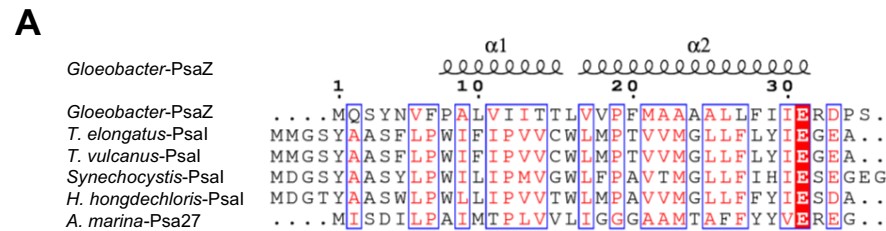

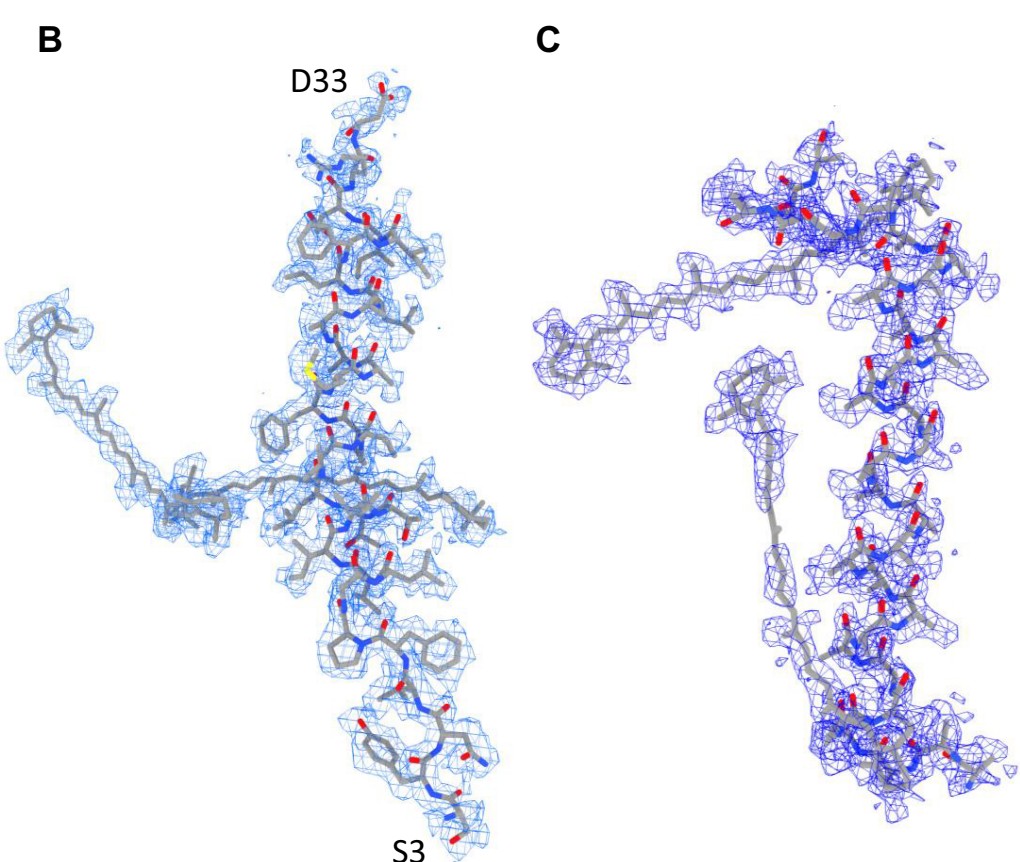

**Appendix 1—figure 3.** Identification of the *Gloeobacter* PSI subunits. (**A**) Multiple sequence alignment (ClustalW and ESPript) of the *Gloeobacter* PsaZ and other cyanobacterial PsaI subunits. The species shown are *Gloeobacter violaceus* PCC 7421, *Thermosynechococcus elongatus* BP-1, *Thermosynechococcus vulcanus* NIES-2134, *Synechocystis* sp. PCC 6803, *Halomicronema hongdechloris* C2206, and *Acaryochloris marina* MBIC11017. (**B**) The cryo-EM density (blue meshes) of PsaZ and its corresponding model (gray sticks). (**C**) The cryo-EM density of the Unknown subunit (blue meshes) and its corresponding model (gray sticks).

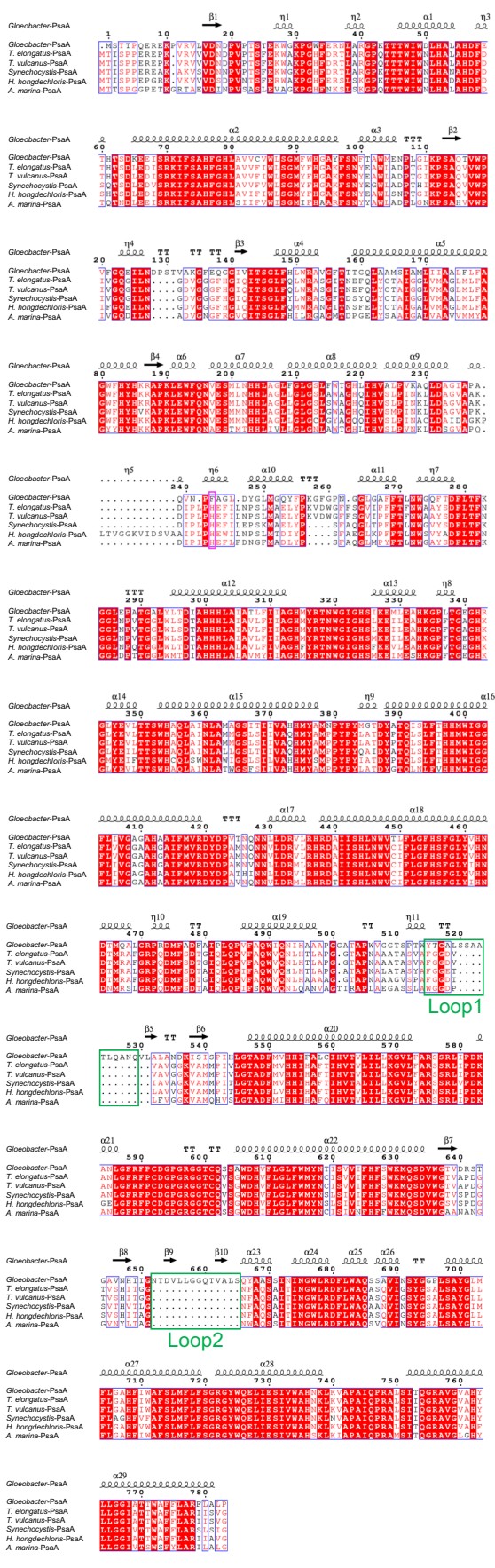

**Appendix 1—figure 4.** Comparison of the structurally known PsaA subunit among six species of cyanobacteria. Multiple sequence alignment was carried out using ClustalW and ESPript. Secondary-structural elements are shown above the sequence. Completely conserved residues are highlighted in red. Loop1 (Tyr515-Gln529) and Loop2 (Asn652-Ser665) are labeled and indicated by green boxes. The pink box stands for the histidine residue involved in the binding of Chl1A. The species shown are *Gloeobacter violaceus* PCC 7421, *Thermosynechococcus elongatus* BP-1, *Thermosynechococcus vulcanus* NIES-2134, *Synechocystis* sp. PCC 6803, *Halomicronema hongdechloris* C2206, and *Acaryochloris marina* MBIC11017.

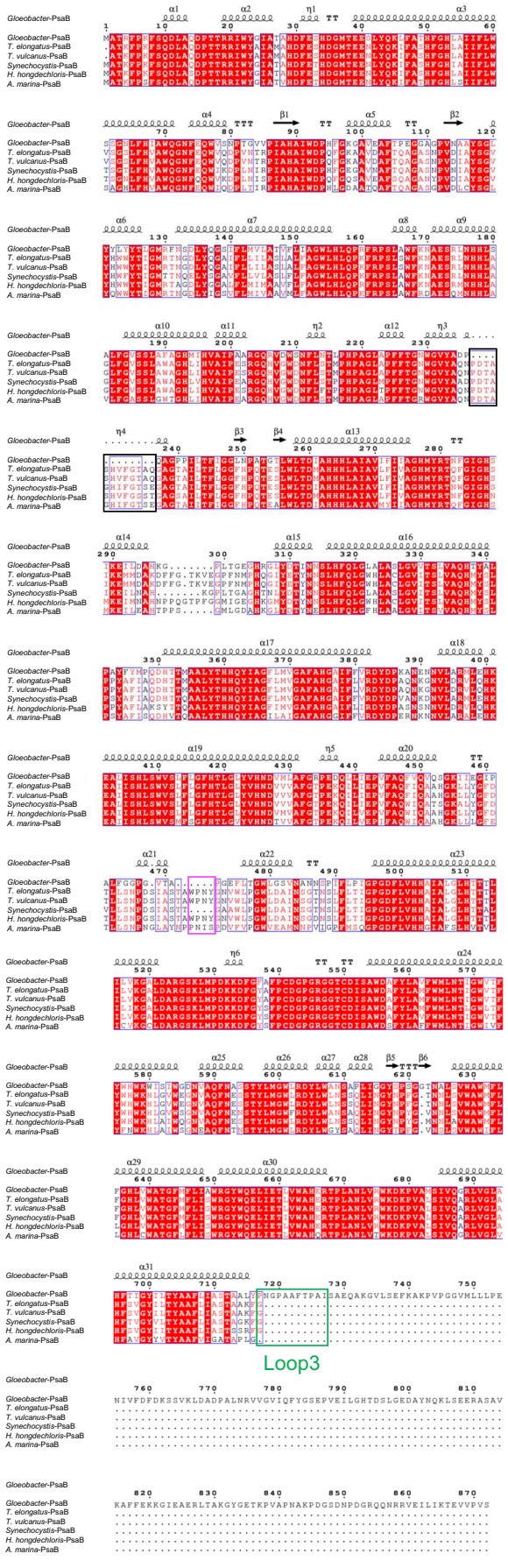

**Appendix 1—figure 5.** Comparison of the structurally known PsaB subunit among six species of cyanobacteria. Multiple sequence alignment was carried out using ClustalW and ESPript. Secondary-structural elements are shown above the sequence. Completely conserved residues are highlighted in red. Loop3 (Pro717-Ile727) is labeled and indicated by a green box. The pink box indicates the loop involved in the binding of Chl1B. The black box stands for the deletion in the *Gloeobacter* PsaB. The species shown are *Gloeobacter violaceus* PCC 7421, *Thermosynechococcus elongatus* BP-1, *Thermosynechococcus vulcanus* NIES-2134, *Synechocystis* sp. PCC 6803, *Halomicronema hongdechloris* C2206, and *Acaryochloris marina* MBIC11017.

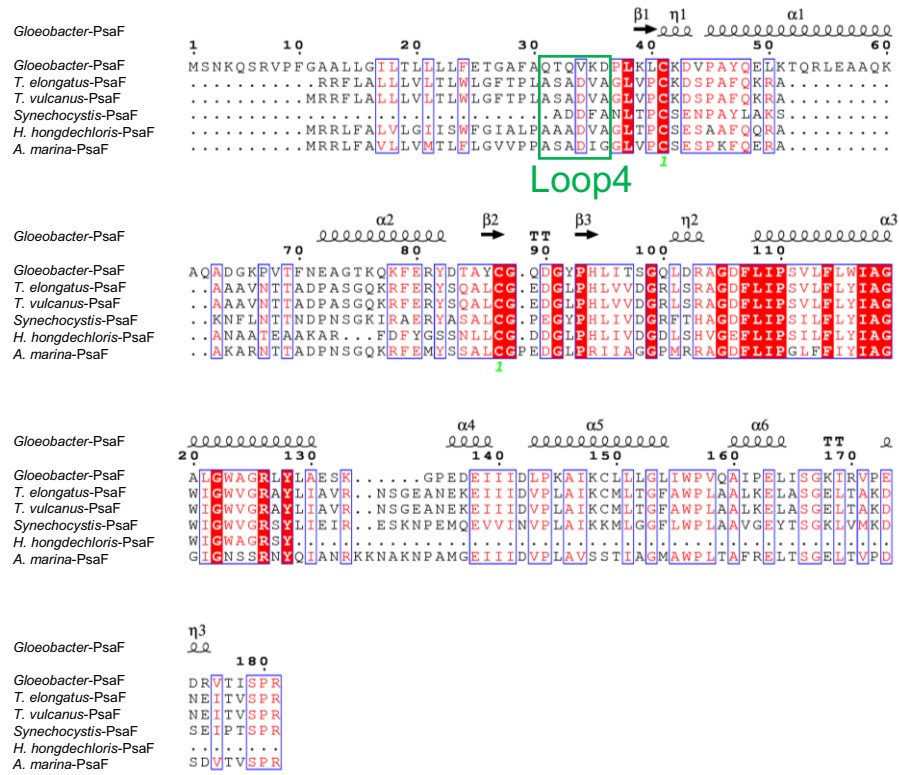

**Appendix 1—figure 6.** Comparison of the structurally known PsaF subunit among six species of cyanobacteria. Multiple sequence alignment was carried out using ClustalW and ESPript. Secondary-structural elements are shown above the sequence. Completely conserved residues are highlighted in red. Loop4 (Gln31-Asp36) is labeled and indicated by a green box. The species shown are *Gloeobacter violaceus* PCC 7421, *Thermosynechococcus elongatus* BP-1, *Thermosynechococcus vulcanus* NIES-2134, *Synechocystis* sp. PCC 6803, *Halomicronema hongdechloris* C2206, and *Acaryochloris marina* MBIC11017.

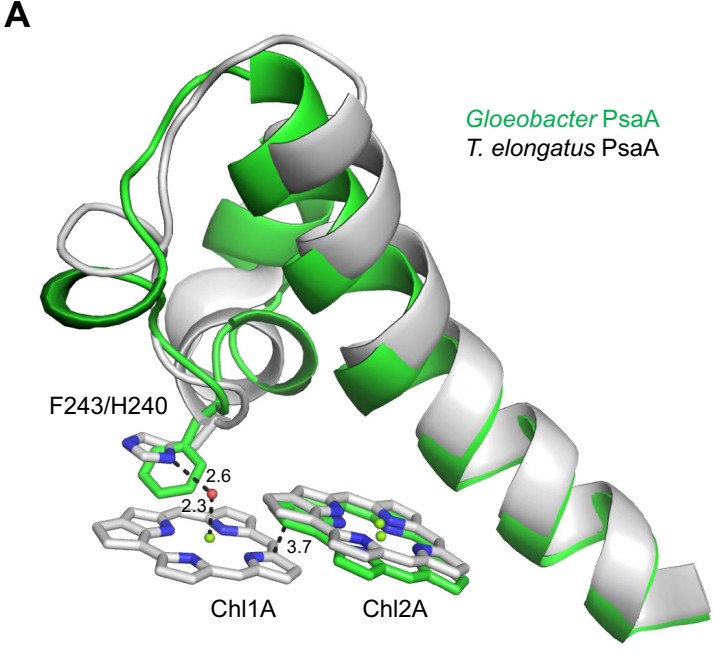

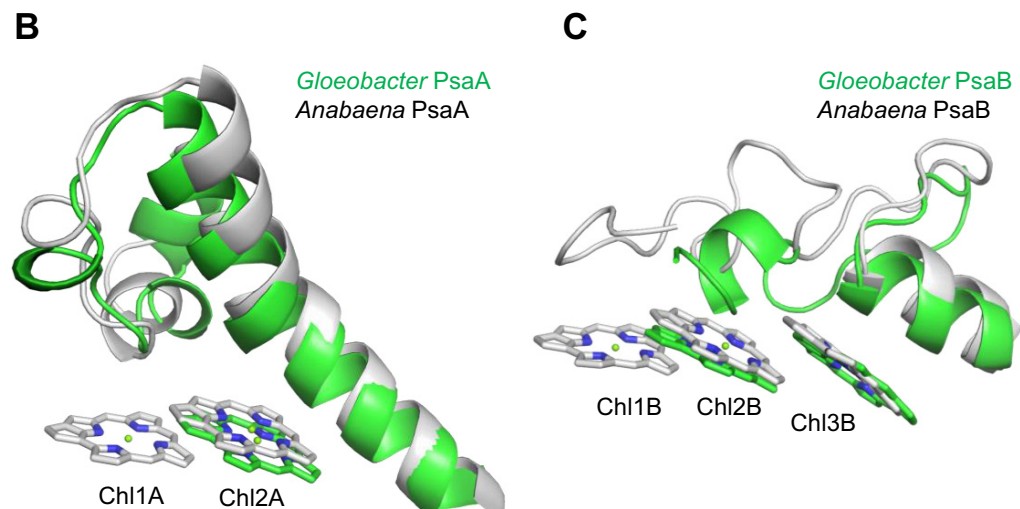

**Appendix 1—figure 7.** Structural comparisons of the Low1 and Low2 sites in PsaA between *Gloeobacter* and other organisms. (**A**) Superposition of the Low1 site in the *Gloeobacter* PSI (green) with that of the *T. elongatus* PSI (gray) (PDB: 1JB0). Phe243 of the *Gloeobacter* PsaA and His240 of the *T. elongatus* PsaA are shown in stick models. Dashed lines indicate interactions with distances labeled in Å. (**B, C**) Superposition of the Low1 (**B**) and Low2 (**C**) sites in the *Gloeobacter* PSI (green) with those in the *Anabaena* PSI (gray) (PDB: 6JEO).

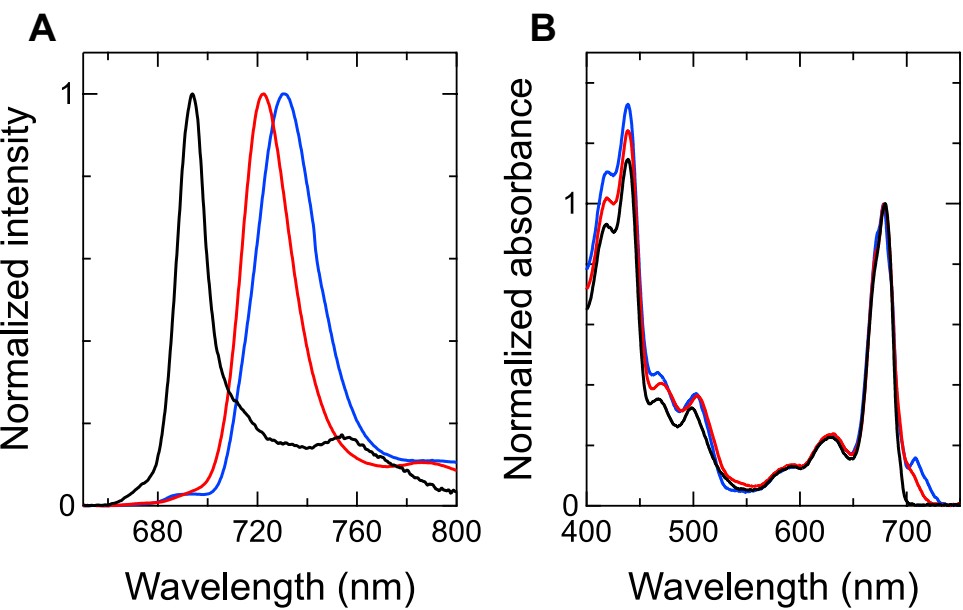

**Appendix 1—figure 8.** Fluorescence-emission and absorption spectra of the *Gloeobacter* PSI (black), *Synechocystis* PSI (red), and *T. vulcanus* PSI (blue). (**A**) Fluorescence spectra were measured at 77 K and normalized by their maximum-peak intensities. (**B**) Absorption spectra were measured at 77 K and normalized by the Qy-peak intensities.

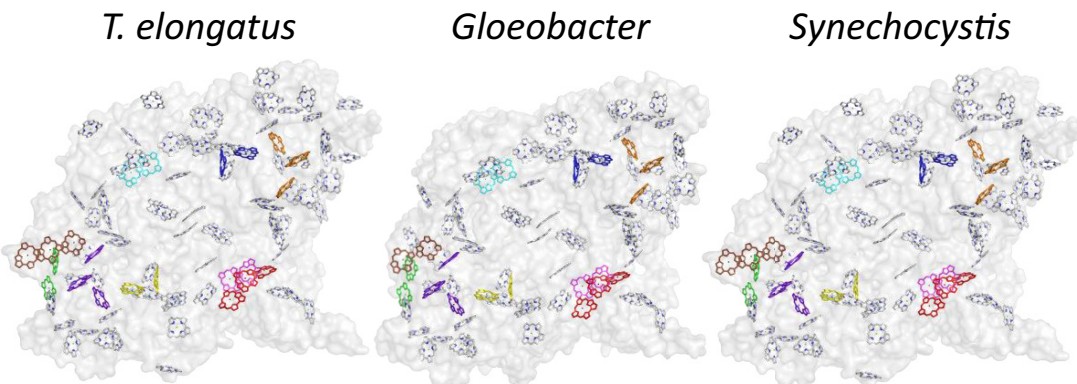

| Chl color | *T. elongatus* (PDB: 1JB0) | *Gloeobacter* (this study) | *Synechocystis* (PDB: 5OY0) |
|---|---|---|---|
| red | A1131/A1132/B1207 | A834/A835/B810 | A1131/A1132/B1207 |
| green | B1218/B1219 | B821/B822 | B1218/B1219 |
| blue | A1126/A1127 | A829/A830 | A1126/A1127 |
| yellow | B1224/B1225 | B827/B828 | B1224/B1225 |
| magenta | A1237/B1238 | A844/B840 | B1237/B1238 |
| cyan | A1138/A1139 | A841/A842 | A1138/A1139 |
| orange | A1116/A1117/A1125 | A819/A820/A828 | A1116/A1117/A1125 |
| purpleblue | B1214/B1215/B1223 | B817/B818/B826 | B1214/B1215/B1223 |
| brown | B1231/B1232/B1233 | B834/B835 | B1231/B1232 |

**Appendix 1—figure 9.** Possible low-energy Chls proposed by *Schlodder et al., 2007*. Nine groups of Chls proposed as low-energy Chls were colored differently in the PSI structures of three cyanobacteria, whereas the other Chls and protein structures were colored gray. The nine groups were listed in the table, with the numbering of these Chls taken from each PDB data.

**A**

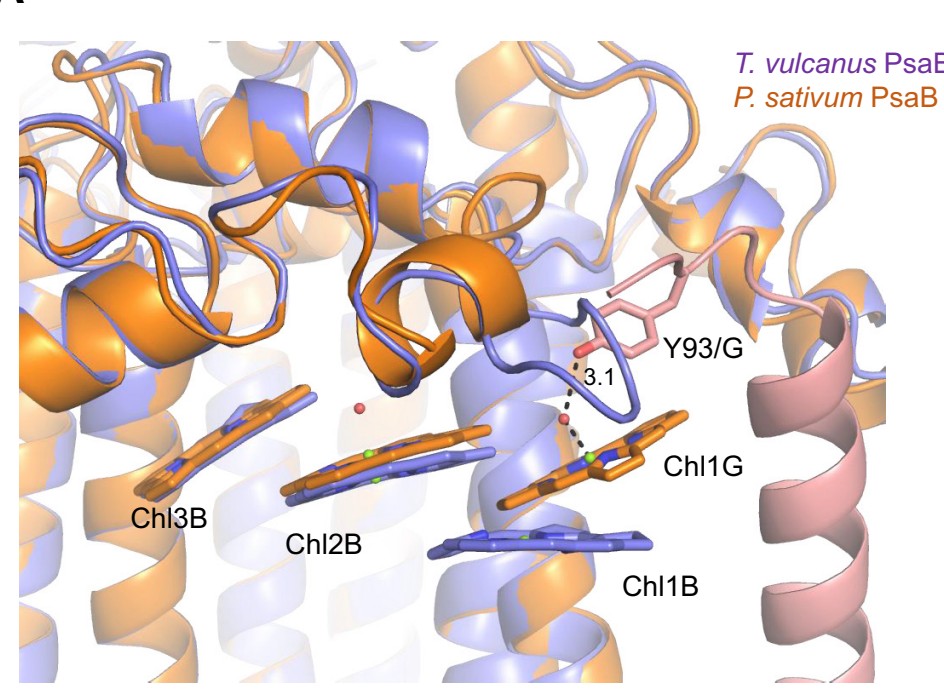

**B**

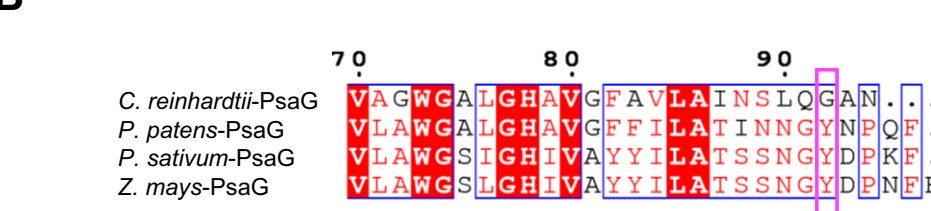

**Appendix 1—figure 10.** Structural comparisons of the Low2 site. (**A**) Superposition of the Low2 site and its surrounding environment between the *T. vulcanus* PsaB (purple) (PDB: 6K33) and the *P. sativum* PsaB (orange) and PsaG (pink) (PDB: 4XK8). Tyr93 of the *P. sativum* PsaG involved in the binding of Chl1G is shown in a stick model. (**B**) Multiple sequence alignment of PsaG using ClustalW and ESPript. The species shown are *Chlamydomonas reinhardtii*, *Physcomitrella patens*, *Pisum sativum*, and *Zea mays*. The pink box displays the tyrosine residue involved in the binding of Chl1G.

**Appendix 1—table 1.** Cryo-EM data collection and structural analysis statistics.

| Complex | PSI trimer |
| --- | --- |
| PDB ID | 7F4V |
| EMDB ID | EMD-31455 |
| Data collection and processing | |
| Magnification | 60,000 |
| Voltage (kV) | 300 |
| Electron exposure (e⁻/Å) | 70.22 |

*Appendix 1—table 1 Continued on next page*

*Appendix 1—table 1 Continued*

| Complex | PSI trimer |
|---|---|
| Defocus range (μm) | −1.8 to −0.6 |
| Pixel size (Å) | 0.823 |
| Symmetry imposed | C3 |
| Initial particle number | 961,960 |
| Final particle number | 261,743 |
| Map resolution (Å) | 2.04 |
| FSC threshold | 0.143 |
| Refinement | |
| Initial Model used (PDB code) | Homology modeling |
| Model resolution (Å) | 2.01 |
| FSC threshold | 0.5 |
| Map sharpening B factor (Å$^2$) | −58.0 |
| Model composition (trimer) | |
| Non-hydrogen atoms | 67,641 |
| Protein | 49,899 |
| Ligand | 17,340 |
| Water | 402 |
| B factors (Å$^2$) | |
| Protein | 39.2 |
| Ligand | 40.5 |
| Water | 27.3 |
| R.m.s deviations | |
| Bond lengths (Å) | 0.025 |
| Bond angles (°) | 2.45 |
| Validation | |
| MolProbity score | 1.78 |
| Clashscore | 4.77 |
| Poor rotamers (%) | 2.78 |
| EMRinger score | 6.48 |
| Ramachandran plot | |
| Favored (%) | 96.8 |
| Allowed (%) | 3.11 |
| Disallowed (%) | 0.09 |

**Appendix 1—table 2.** Averaged *Q*-score in each subunit and cofactors in each monomer unit of the PSI core.

| Protein | Amino acid residues | Averaged Q-score | Chlorophylls | Carotenoids | Lipids | Others |
|---|---|---|---|---|---|---|
| PsaA | 11–782 | 0.85 | 43 Chl *a* 1 Chl *a'* | 5 BCR | 2 LHG | 1 [4Fe-4S] cluster, 1 menaquinone-4 |
| PsaB | 3–727 | 0.87 | 41 Chl *a* | 7 BCR | 1 LMG 1 LHG | one menaquinone-4 |

*Appendix 1—table 2 Continued on next page*

*Appendix 1—table 2 Continued*

| Protein | Amino acid residues | Averaged Q-score | Chlorophylls | Carotenoids | Lipids | Others |
|---------|---------------------|------------------|--------------|-------------|--------|--------|
| PsaC | 2–81 | 0.87 | - | - | - | 2 [4Fe-4S] cluster |
| PsaD | 11–143 | 0.81 | - | - | - | - |
| PsaE | 3–64 | 0.76 | - | - | - | - |
| PsaF | 31–181 | 0.76 | 1 Chl *a* | 1 BCR | - | - |
| PsaZ | 3–33 | 0.85 | - | 2 BCR | - | - |
| Unknown | 1–33 | 0.80 | - | 2 BCR | - | - |
| PsaL | 11–782 | 0.85 | 3 Chl *a* | 2 BCR | - | - |
| PsaM | 3–727 | 0.84 | - | 1 BCR | - | - |
| Total | | | 89 | 20 | 4 | 5 |

