## [Editor Report]

This work presents a high-resolution structure of a photosystem I complex of the primordial cyanobacterium *Gloeobacter violaceus*. It relates the structural differences with oher cyanobacteria to differences in the optical properties.

---

## [Decision Letter]

**Decision letter after peer review:**

Thank you for submitting your article "Structural basis for the absence of low-energy chlorophylls responsible for photoprotection from a primitive cyanobacterial PSI" for consideration by *eLife*. Your article has been reviewed by 3 peer reviewers, and the evaluation has been overseen by a Reviewing Editor and Jürgen Kleine-Vehn as the Senior Editor. The following individual involved in review of your submission has agreed to reveal their identity: Thomas Renger (Reviewer #1).

Essential revisions:

The reviewers agree that your identification of red Chls in PSI is easily spectacular enough to justify publication in *eLife*. In contrast your discussion of possible roles for these chlorophylls is not supported by any data.

1) We ask you to rewrite the paper to take out the non-supported statements and conclusions (please see reviewers' comments). The reviewers think you can write a nice paper focusing on the comparison of the structures and the location of the red forms including a discussion of the literature. You need to also discuss the limitations of the assignment. PSI photoprotection is not well established, so since the you have no evidence for it, you should avoid discussing it.

*Reviewer #1 (Recommendations for the authors):*

I have two suggestions for an improvement of the paper:

1) The authors should discuss all possible candidates for red Chls in PSI identified previously. In my view, the best assignment so far was achieved by Schlodder and coworkers that analyzed the linear dichroism of the red states and compared that with an exciton model of the dimers and trimerswith short intermolecular distances of PSI : Schlodder et al., Biochimica et Biophysica Acta 1767 (2007) 732-741. This paper would be worth discussing. On page 735, right column, 9 candidates for red Chls (5 dimers and 4 trimers) are discussed. It would be worth to compare not only the dimers and trimers that have a missing Chl in Gloeobacter, but also the remaining dimers and trimers. In principle, it could be that there are differences in the other dimers and trimers that could also lead to changes in the optical properties. I think the present interpretation is the most straighforward one, but for completeness one should also check the remaining candidates.

2) A discussion is needed about what is known about photoprotection in cyanobacteria, e.g. to what extend does it occur in PSI, PSII or in more peripheral antennae. The reader need to get an impression how likely it is that the missing Chl in PSI of Gloeobacter is responsible for the photodamage that this cynobacterium suffers at high light intensities. In other words, can the authors exclude that there are other differences outside of PSI between Gloeobacter and other cyanobacteria that could be responsible for the better photoprotection of the latter.

*Reviewer #2 (Recommendations for the authors):*

The conclusions of this paper are mostly well supported by the data, which are professionally presented. Two aspects need to be clarified and extended:

(1) When describing the position of Low1 and Low2, the author stated that "However, the interface-type low-energy Chls (in Anabaena PSI tetramer) are different from the Low1 and Low2 observed here." It is better to clarify what is the difference and extend the statement that "the Low1 and Low2 are found to be within each monomer unit of both PSI trimer and tetramer, but not in the interfaces among the PSI monomers." to make it more understandable to the readers.

(2) Figure 2A shows the arrangement of the cofactors involved in the electron-transfer reaction of PSI. It is recommended to describe the electron-transfer path in the text or represent the electron-transfer route between these Chls, menaquinone-4 and iron-sulfur clusters in the figure.

The data availability, code and reagents or other issues are properly presented in the manuscript.

*Reviewer #3 (Recommendations for the authors):*

My suggestion is to avoid the discussion of the physiological implication of the differences and focus on the structural aspects. The localization of the red forms of PSI is a topic that was addressed by many authors using different experimental and theoretical approaches. It would be very helpful if the authors would discuss the literature and compare their suggestions to the previous ones.

The writing needs improvements.

---

## [Author Response]

Reviewer #1 (Recommendations for the authors):I have two suggestions for an improvement of the paper:1) The authors should discuss all possible candidates for red Chls in PSI identified previously. In my view, the best assignment so far was achieved by Schlodder and coworkers that analyzed the linear dichroism of the red states and compared that with an exciton model of the dimers and trimerswith short intermolecular distances of PSI : Schlodder et al., Biochimica et Biophysica Acta 1767 (2007) 732-741. This paper would be worth discussing. On page 735, right column, 9 candidates for red Chls (5 dimers and 4 trimers) are discussed. It would be worth to compare not only the dimers and trimers that have a missing Chl in Gloeobacter, but also the remaining dimers and trimers. In principle, it could be that there are differences in the other dimers and trimers that could also lead to changes in the optical properties. I think the present interpretation is the most straighforward one, but for completeness one should also check the remaining candidates.

We fully agree with the reviewer’s comments. According to your comments, we cited the paper of Schlodder et al., and added a new Figure (Figure S9) to the revised manuscript and compared all the 9 candidates listed in the reference with those identified in the present study as well as in other cyanobacteria. Among them, Low2 was found to be different in the Gloeobacter and other cyanobacteria; however, Low1 was not listed as a candidate in the Schlodder et al’s research. We discussed the possibility of failure of Low1 assignment by the spectroscopic method used in the reference. These contents were added to the section of “Correlation of Low1 and Low2 with fluorescence spectra” in the revised manuscript (from line 15 in page 9 to line 9 in page 10).

2) A discussion is needed about what is known about photoprotection in cyanobacteria, e.g. to what extend does it occur in PSI, PSII or in more peripheral antennae. The reader need to get an impression how likely it is that the missing Chl in PSI of Gloeobacter is responsible for the photodamage that this cynobacterium suffers at high light intensities. In other words, can the authors exclude that there are other differences outside of PSI between Gloeobacter and other cyanobacteria that could be responsible for the better photoprotection of the latter.

We agree with the reviewer’s comments that the extent that these low-energy Chls may contribute to the PSI photoprotection is unknown. In view of the comments and suggestions of other reviewers, we removed all sentences regarding photoprotection from the revised manuscript, in order to avoid misleading.

Reviewer #2 (Recommendations for the authors):The conclusions of this paper are mostly well supported by the data, which are professionally presented. Two aspects need to be clarified and extended:(1) When describing the position of Low1 and Low2, the author stated that "However, the interface-type low-energy Chls (in Anabaena PSI tetramer) are different from the Low1 and Low2 observed here." It is better to clarify what is the difference and extend the statement that "the Low1 and Low2 are found to be within each monomer unit of both PSI trimer and tetramer, but not in the interfaces among the PSI monomers." to make it more understandable to the readers.

According to your comment, we largely improved the paragraph in the revised manuscript. The content of *Anabaena* was added to the section of “Correlation of Low1 and Low2 with fluorescence spectra”. Also, the section of “Conservation of Low1 and Low2 in PSI-monomer units but not in their interfaces” was changed to “Low-energy Chls in the interface among PSI-monomer units”, where the explanation regarding the differences between Low1/Low2 and low-energy Chls located in the interfaces were added (page 10).

(2) Figure 2A shows the arrangement of the cofactors involved in the electron-transfer reaction of PSI. It is recommended to describe the electron-transfer path in the text or represent the electron-transfer route between these Chls, menaquinone-4 and iron-sulfur clusters in the figure.

According to your comments, we added the information of electron transfer to the Figure legend of Figure 2 in the revised manuscript. The electron transfer mechanisms are still under debate, e.g., which branch(es) is(are) used for electron transfer is still not clear, and whether Acc functions as an electron donor is also a matter of debate. Therefore, to avoid misleadings, we do not add the electron flow as arrows to Figure 2A in the revised manuscript. Instead, we added sentences to indicate this in the revised text (lines 25-29, Page 5) and the legend of Figure 2 (page 22).

The data availability, code and reagents or other issues are properly presented in the manuscript.

We have revised the data availability, code and reagents, etc., to make them clearer.

Reviewer #3 (Recommendations for the authors):My suggestion is to avoid the discussion of the physiological implication of the differences and focus on the structural aspects. The localization of the red forms of PSI is a topic that was addressed by many authors using different experimental and theoretical approaches. It would be very helpful if the authors would discuss the literature and compare their suggestions to the previous ones.

We fully agree with the reviewer’s comments, and have focused our results on the structural aspects, removed all sentences regarding the physiological implications of Low1 and Low2. We also discussed our suggestions in relation to the previous ones in the literature.

The writing needs improvements.

We have looked over the entire text and improved the writing where necessary.